# Beyond Redundancy: Information-aware Unsupervised Multiplex Graph Structure Learning

**Zhixiang Shen**,[*] **Shuo Wang**,[*] **Zhao Kang**[†]
School of Computer Science and Engineering,
University of Electronic Science and Technology of China, Chengdu, Sichuan, China
zhixiang.zxs@gmail.com  zkang@uestc.edu.cn

## Abstract

Unsupervised Multiplex Graph Learning (UMGL) aims to learn node representations on various edge types without manual labeling. However, existing research overlooks a key factor: the reliability of the graph structure. Real-world data often exhibit a complex nature and contain abundant task-irrelevant noise, severely compromising UMGL's performance. Moreover, existing methods primarily rely on contrastive learning to maximize mutual information across different graphs, limiting them to multiplex graph redundant scenarios and failing to capture view-unique task-relevant information. In this paper, we focus on a more realistic and challenging task: to unsupervisedly learn a fused graph from multiple graphs that preserve sufficient task-relevant information while removing task-irrelevant noise. Specifically, our proposed **Info**rmation-aware Unsupervised **M**ultiplex **G**raph **F**usion framework (InfoMGF) uses graph structure refinement to eliminate irrelevant noise and simultaneously maximizes view-shared and view-unique task-relevant information, thereby tackling the frontier of non-redundant multiplex graph. Theoretical analyses further guarantee the effectiveness of InfoMGF. Comprehensive experiments against various baselines on different downstream tasks demonstrate its superior performance and robustness. Surprisingly, our unsupervised method even beats the sophisticated supervised approaches. The source code and datasets are available at https://github.com/zxlearningdeep/InfoMGF.

## 1  Introduction

Multiplex graph (multiple graph layers span across a common set of nodes), as a special type of heterogeneous graph, provides richer information and better modeling capabilities, leading to challenges in learning graph representation [1]. Recently, unsupervised multiplex graph learning (UMGL) has attracted significant attention due to its exploitation of more detailed information from diverse sources [2, 3], using graph neural networks (GNNs) [4] and self-supervised techniques [5]. UMGL has become a powerful tool in numerous real-world applications [6, 7], e.g., social network mining and biological network analysis, where multiple relationship types exist or various interaction types occur.

Despite the significant progress made by UMGL, a substantial gap in understanding how to take advantage of the richness of the multiplex view is still left. In particular, a fundamental issue is largely overlooked: the reliability of graph structure. Typically, the messaging-passing mechanism in GNNs assumes the reliability of the graph structure, implying that the connected nodes tend to have similar labels. All UMGL methods are graph-fixed, assuming that the original structure is sufficiently reliable for learning [3, 8–10]. Unfortunately, there has been evidence that practical

---

[*]Equal contribution.
[†]Corresponding author.

38th Conference on Neural Information Processing Systems (NeurIPS 2024).

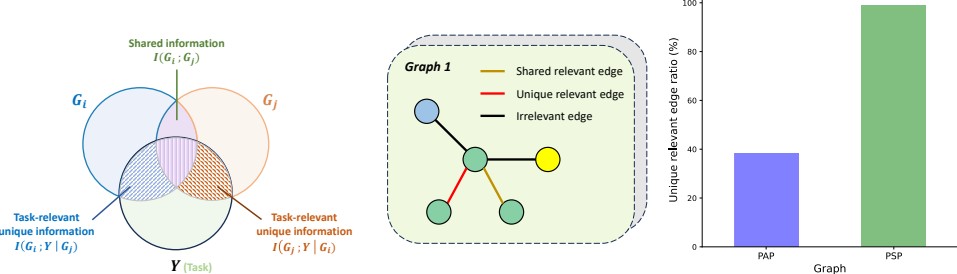

(a) Multiplex graph non-redundancy    (b) Non-redundancy example    (c) Empirical study on ACM

Figure 1: (a) and (b) illustrate that in a non-redundant multiplex graph, view-specific task-relevant edges exist in certain graphs. The color of nodes represents class, edges between nodes of the same class are considered relevant edges, and "unique" indicates that the edge exists only in one graph. (c) The unique relevant edge ratio = (the number of unique relevant edges) / (the total number of relevant edges in this graph). Each graph contains a significant amount of unique task-relevant information.

graph structures are not always reliable [11]. Multiplex graphs often contain substantial amounts of less informative edges characterized by irrelevant, misleading, and missing connections. For example, due to the heterophily in the graphs, GNNs generate poor performance [12–14]. Another representative example is adversarial attacks [15], where attackers tend to add edges between nodes of different classes. Then, aggregating information from neighbors of different classes degrades UMGL performance. Diverging from existing approaches to node representation learning, we focus on structure learning of a new graph from multiplex graphs to better suit downstream tasks. Notably, existing Graph Structure Learning (GSL) overwhelmingly concentrated on a single homogeneous graph [16], marking our endeavor as pioneering in the realm of multiplex graphs.

Given the unsupervised nature, the majority of UMGL methods leverage contrastive learning mechanism [8–10], a typical self-supervised technique, for effective training. However, recent research has demonstrated that standard contrastive learning, maximizing mutual information between different views, is limited to capturing view-shared task-relevant information [17]. This approach is effective only in multi-view redundant scenarios, thereby overlooking unique task-relevant information specific to each view. In practice, the multiplex graph is inherently non-redundant. As illustrated in Figure 1, task-relevant information resides not only in shared areas across different graph views but also in specific view-unique regions. For instance, in the real citation network ACM [18], certain papers on the same subject authored by different researchers may share categories and thematic relevance. This characteristic, compared to the co-author view, represents view-unique task-relevant information within the co-subject view. It exposes a critical limitation in existing UMGL methods, which potentially cannot capture sufficient task-relevant information.

Motivated by the above observations, our research goal can be summarized as follows: ***how can we learn a fused graph from the original multiplex graph in an unsupervised manner, mitigating task-irrelevant noise while retaining sufficient task-relevant information?*** To handle this new task, we propose a novel Information-aware Unsupervised Multiplex Graph Fusion framework (InfoMGF). Graph structure refinement is first applied to each view to achieve a more suitable graph with less task-irrelevant noise. Confronting multiplex graph non-redundancy, InfoMGF simultaneously maximizes the view-shared and view-unique task-relevant information to realize sufficient graph learning. A learnable graph augmentation generator is also developed. Finally, InfoMGF maximizes the mutual information between the fused graph and each refined graph to encapsulate clean and holistic task-relevant information from a range of various interaction types. Theoretical analyses guarantee the effectiveness of our approach in capturing task-relevant information and graph fusion. The unsupervised learned graph and node representations can be applied to various downstream tasks. In summary, our main contributions are three-fold:

- **Problem.** We pioneer the investigation of the multiplex graph reliability problem in a principled way, which is a more practical and challenging task. To our best knowledge, we are the first to attempt unsupervised graph structure learning in multiplex graphs.

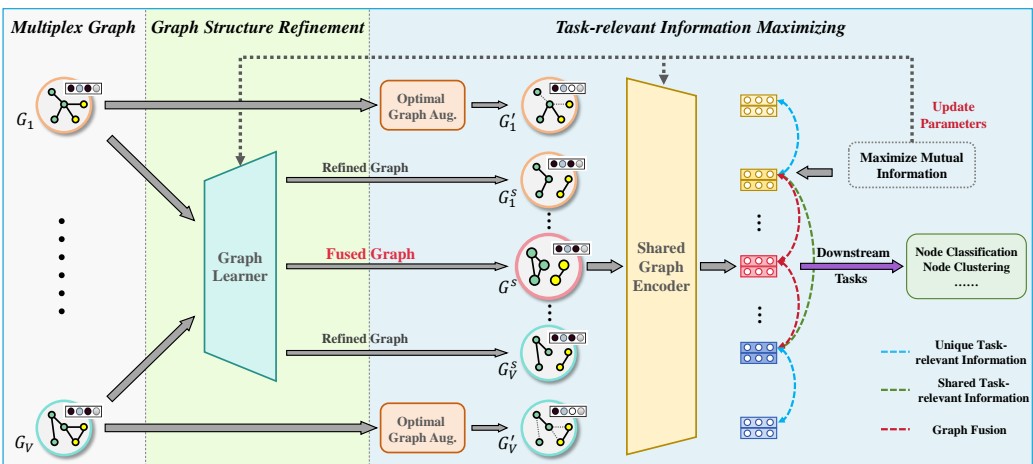

Figure 2: The overall framework of the proposed InfoMGF. Specifically, InfoMGF first generates refined graphs and the fused graph through the graph learner. Subsequently, it maximizes shared and unique task-relevant information within the multiplex graph and facilitates graph fusion. The learned fused graph and node representations are used for various downstream tasks.

- **Algorithm.** We propose InfoMGF, a versatile multiplex graph fusion framework that steers the fused graph learning by concurrently maximizing both view-shared and view-unique task-relevant information under the multiple graphs non-redundancy principle. Furthermore, we develop two random and generative graph augmentation strategies to capture view-unique task information. Theoretical analyses ensure the effectiveness of InfoMGF.

- **Evaluation.** We perform extensive experiments against various types of state-of-the-art methods on different downstream tasks to comprehensively evaluate the effectiveness and robustness of InfoMGF. Particularly, our developed unsupervised approach even outperforms supervised methods.

## 2 Preliminaries

**Notation.** The multiplex graph is represented by $G = \{G_1, ..., G_V\}$, where $G_v = \{A_v, X\}$ is the $v$-th graph. $A_v \in \{0,1\}^{N \times N}$ is the corresponding adjacency matrix and $X \in \mathbb{R}^{N \times d_f}$ is the shared feature matrix across all graphs. $X_i \in \mathbb{R}^{d_f}$ is the $i$-th row of $X$, representing the feature vector of node $i$. $N$ is the number of nodes and $D_v$ is a diagonal matrix denoting the degree matrix of $A_v$. $Y$ is label information. For convenience, we use "view" to refer to each graph in the multiplex graph.

**Multiplex graph non-redundancy.** Task-relevant information exists not only in the shared information between graphs but also potentially within the unique information of certain graphs. Following the non-redundancy principle [17], we provide the formal definition of Multiplex Graph Non-redundancy:

**Definition 1.** *$G_i$ is considered non-redundant with $G_j$ for $Y$ if and only if there exists $\epsilon > 0$ such that the conditional mutual information $I(G_i; Y \mid G_j) > \epsilon$ or $I(G_j; Y \mid G_i) > \epsilon$.*

**Graph structure learning.** Existing GSL methods primarily focus on a single graph. Their pipeline can be summarized as a two-stage framework [16]: a Graph Learner takes in the original graph $G = \{A, X\}$ to generate a refined graph $G^s = \{A^s, X\}$ with a new structure; a Graph Encoder uses the refined graph as input to obtain node representations. Note that node features generally do not change in GSL, only the graph structure is optimized. Related work is in Appendix B.

## 3 Methodology

As illustrated in Figure 2, our proposed InfoMGF consists of two modules: the *Graph Structure Refinement module* and the *Task-Relevant Information Maximization module*.

## 3.1 Graph Structure Refinement

We first use a graph learner to generate each view's refined graph $G_v^s = \{A_v^s, X\}$. To retain node features and structure information simultaneously, we apply the widely used Simple Graph Convolution (SGC) [19] to perform aggregation in each view, resulting in view-specific node features $X^v$. A view-specific two-layer attentive network is employed to model the varying contributions of different features to structure learning:

$$X^v = (\tilde{D}_v^{-\frac{1}{2}} \tilde{A}_v \tilde{D}_v^{-\frac{1}{2}})^r X, \quad H^v = \sigma(X^v \odot W_1^v) \odot W_2^v \tag{1}$$

where $\tilde{D}_v = D_v + I$ and $\tilde{A}_v = A_v + I$. $r$ represents the order of graph aggregation. $\sigma(\cdot)$ is the non-linear activation function and $\odot$ denotes the Hadamard product. All rows of $W_1^v$ are identical, representing a learnable attention vector shared by all nodes. This strategy enables us to acquire view-specific features before training, thereby circumventing the time-consuming graph convolution operations typically required by GNN-based graph learners during training, which significantly boosts our model's scalability.

Like existing GSL methods [16, 20], we apply post-processing techniques to ensure that the adjacency matrix $A_v^s$ satisfies properties such as sparsity, non-negativity, symmetry, and normalization. Specifically, we use $H^v$ to construct the similarity matrix and then sparsify it using $k$-nearest neighbors ($k$NN). For large-scale graphs, we utilize locality-sensitive approximation during $k$NN sparsification to reduce time complexity [21]. Afterward, operations including Symmetrization, Activation, and Normalization are used sequentially to generate the final $A_v^s$. Following the refinement of each view, we employ a shared Graph Convolutional Network (GCN) [22] as the graph encoder to obtain the node representations $Z^v \in \mathbb{R}^{N \times d}$ of each view, computed by $Z^v = \text{GCN}(A_v^s, X)$.

## 3.2 Maximizing Shared Task-Relevant Information

$G_v^s$ should contain not only view-shared but also view-unique task-relevant information. Following standard contrastive learning [23, 24], for each pair of distinct views (e.g., $i$ and $j$), our approach seeks to maximize the mutual information $0.5I(G_i^s; G_j) + 0.5I(G_j^s; G_i)$ to capture shared task-relevant information between views.

**Proposition 1.** *For any view $i$ and $j$, $2I(G_i^s; G_j^s)$ is the lower bound of $I(G_i^s; G_j) + I(G_j^s; G_i)$.*

Detailed proofs are provided in the Appendix D. According to Proposition 1, the maximization objective can be transformed to a tractable lower bound $I(G_i^s; G_j^s)$. Considering the addition of mutual information for each pair, the loss term for minimization can be expressed as follows:

$$\mathcal{L}_s = -\frac{2}{V(V-1)} \sum_{i=1}^{V} \sum_{j=i+1}^{V} I(G_i^s; G_j^s) \tag{2}$$

## 3.3 Maximizing Unique Task-Relevant Information

Maximizing view-unique task-relevant information can be rigorously expressed as maximizing $I(G_i^s; Y \mid \cup_{j \neq i} G_j)$. Then, we relax the optimization objective to the total task-relevant information within the view, $I(G_i^s; Y)$. This decision is based on the following considerations: on the one hand, deliberately excluding shared task-relevant information is unnecessary and would complicate the optimization process. On the other hand, repeated emphasis on shared task-relevant information encourages the model to focus more on it in the early training stage.

The unsupervised nature of our task dictates that we cannot directly optimize $I(G_i^s; Y)$ using label information. Some typical graph learning methods often reconstruct the graph structure to preserve the maximum amount of information from the original data [25–27]. In the context of our task, this reconstruction-based optimization objective is equivalent to maximizing the mutual information with the original graph structure [28, 29], i.e., $I(G_i^s; G_i)$. However, such methods have significant drawbacks: they retain task-irrelevant information from the original data, and the graph reconstruction also entails high complexity. In contrast, we leverage graph augmentation to reduce task-irrelevant information and retain task-relevant information without accessing $Y$. Following the optimal augmentation assumption [17, 30], we define optimal graph augmentation as:

**Definition 2.** $G_i'$ is an optimal augmented graph of $G_i$ if and only if $I(G_i'; G_i) = I(Y; G_i)$, implying that the only information shared between $G_i$ and $G_i'$ is task-relevant without task-irrelevant noise.

**Theorem 1.** If $G_i'$ is the optimal augmented graph of $G_i$, then $I(G_i^s; G_i') = I(G_i^s; Y)$ holds.

**Theorem 2.** The maximization of $I(G_i^s; G_i')$ yields a discernible reduction in the task-irrelevant information relative to the maximization of $I(G_i^s; G_i)$.

Theorem 1 theoretically guarantees that maximizing $I(G_i^s; G_i')$ would provide clean and sufficient task-relevant guidance for learning $G_i^s$. Theorem 2 demonstrates the superiority of our optimization objective over typical methods in removing task-irrelevant information. Therefore, given $G_i' = \{A_i', X'\}$ for each view, where $A_i'$ and $X'$ denote the augmented adjacency matrix and node features, respectively, the loss term $\mathcal{L}_u$ is defined as:

$$\mathcal{L}_u = -\frac{1}{V} \sum_{i=1}^{V} I(G_i^s; G_i') \tag{3}$$

The key to the above objective lies in ensuring that $G_i'$ satisfies the optimal graph augmentation. However, given the absence of label information, achieving truly optimal augmentation is not feasible; instead, we can only rely on heuristic techniques to simulate it. Consistent with most existing graph augmentations, we believe that task-relevant information in graph data exists in both structure and feature, necessitating augmentation in both aspects. We use random masking, a simple yet effective method, to perform feature augmentation. For graph structure, we propose two versions: random edge dropping and learnable augmentation through a graph generator.

**Random feature masking.** For node features, we randomly select a fraction of feature dimensions and mask them with zeros. Formally, we sample a random vector $\vec{m} \in \{0,1\}^{d_f}$ where each dimension is drawn from a Bernoulli distribution independently, i.e., $\vec{m}_i \sim Bern(1 - \rho)$. Then, the augmented node features $X'$ is computed by $X' = [X_1 \odot \vec{m}; X_2 \odot \vec{m}; ...; X_N \odot \vec{m}]^\top$.

**Random edge dropping (InfoMGF-RA).** For a given $A_v$, a masking matrix $M \in \{0,1\}^{N \times N}$ is randomly generated, where each element $M_{ij}$ is sampled from a Bernoulli distribution. Afterward, the augmented adjacency matrix can be computed as $A_v' = A_v \odot M$.

**Learnable generative augmentation (InfoMGF-LA).** Random edge dropping may lack reliability and interpretability. A low dropping probability might not suffice to eliminate task-irrelevant information, while excessive deletions could compromise task-relevant information. Therefore, we opt to use a learnable graph augmentation generator. To avoid interference from inappropriate structure information, we compute personalized sampling probabilities for existing edges in each view by employing a Multilayer Perceptron (MLP) in the node features. To ensure the differentiability of the sampling operation for end-to-end training, we introduce the Gumbel-Max reparametrization trick [31, 32] to transform the discrete binary (0-1) distribution of edge weights into a continuous distribution. Specifically, for each edge $e_{i,j}$ in view $v$, its edge weight $\omega_{i,j}^v$ in the corresponding augmented view is computed as follows:

$$\theta_{i,j}^v = \text{MLP}\left([W X_i; W X_j]\right), \quad \omega_{i,j}^v = \text{Sigmoid}\left((\log \delta - \log(1 - \delta) + \theta_{i,j}^v)/\tau\right) \tag{4}$$

where $[\cdot; \cdot]$ denotes the concatenation operation and $\delta \sim \text{Uniform}(0,1)$ is the sampled Gumbel random variate. We can control the temperature hyper-parameter $\tau$ approaching 0 to make $\omega_{i,j}^v$ tend towards a binary distribution. For an effective augmented graph generator, it should eliminate task-irrelevant noise while retaining task-relevant information. Therefore, we design a suitable loss function for augmented graph training:

$$\mathcal{L}_{gen} = \frac{1}{NV} \sum_{i=1}^{V} \sum_{j=1}^{N} \left(1 - \frac{(X_j^i)^\top \hat{X}_j^i}{\|X_j^i\| \cdot \|\hat{X}_j^i\|}\right) + \lambda * \frac{1}{V} \sum_{i=1}^{V} I(G_i^s; G_i') \tag{5}$$

where $\lambda$ is a positive hyper-parameter. The first term reconstructs view-specific features using the cosine error, guaranteeing that the augmented views preserve crucial task-relevant information while having lower complexity compared to reconstructing the entire graph structure. The reconstructed features $\hat{X}^i$ are obtained using an MLP-based Decoder on the node representations $Z^{i'}$ of the augmented view. The second term **minimizes** $I(G_i^s; G_i')$ to regularize the augmented views simultaneously, ensuring that the augmented graphs would provide only task-relevant information as guidance

with less task-irrelevant noise when optimizing the refined graph $G_i^s$ through Eq.(3). Note that for InfoMGF-LA, we adopt an iterative optimization strategy to update $G_i^s$ and $G_i'$ alternatively, as described in Section 3.4.

Although previous work also employs similar generative graph augmentation [33], we still possess irreplaceable advantages in comparison. Firstly, they merely minimize mutual information to generate the augmented graph, lacking the crucial information retention component, which may jeopardize task-relevant information. Furthermore, an upper bound should ideally be used for minimization, whereas they utilize a lower bound estimator for computation, which is incorrect in optimization practice. In contrast, we use a rigorous upper bound of mutual information for the second term of $\mathcal{L}_{gen}$, which is demonstrated later.

### 3.4 Multiplex Graph Fusion

The refined graph retains task-relevant information from each view while eliminating task-irrelevant noise. Afterward, we learn a fused graph that encapsulates sufficient task-relevant information from all views. Consistent with the approach in Section 3.1, we leverage a scalable attention mechanism as the fused graph learner:

$$H = \sigma([X; X^1; X^2; \cdots; X^V] \odot W^1) \odot W^2, \quad \mathcal{L}_f = -\frac{1}{V} \sum_{i=1}^{V} I(G^s; G_i^s) \qquad (6)$$

where the node features are concatenated with all view-specific features as input. The same post-processing techniques are sequentially applied to generate the fused graph $G^s = \{A^s, X\}$. The node representations $Z$ of the fused graph are also obtained through the same GCN. We maximize the mutual information between the fused graph and each refined graph to incorporate task-relevant information from all views, denoted as loss $\mathcal{L}_f$. The total loss $\mathcal{L}$ of our model can be expressed as the sum of three terms: $\mathcal{L} = \mathcal{L}_s + \mathcal{L}_u + \mathcal{L}_f$.

**Theorem 3.** *The learned fused graph $G^s$ contains more task-relevant information than the refined graph $G_i^s$ from any single view. Formally, we have:*

$$I(G^s; Y) \geq \max_i I(G_i^s; Y) \qquad (7)$$

Theorem 3 theoretically proves that the fused graph $G^s$ can incorporate more task-relevant information than considering each view individually, thus ensuring the effectiveness of multiplex graph fusion.

**Optimization.** Note that all the loss terms require calculating mutual information. However, directly computing mutual information between two graphs is impractical due to the complexity of graph-structured data. Since we focus on node-level tasks, we assume the optimized graph should guarantee that each node's neighborhood substructure contains sufficient task-relevant information. Therefore, this requirement can be transferred into mutual information between node representations [34], which can be easily computed using a sample-based differentiable lower/upper bound. For any view $i$ and $j$, the lower bound $I_{lb}$ and upper bound $I_{ub}$ of the mutual information $I(Z^i; Z^j)$ are [17]:

$$I_{lb}(Z^i; Z^j) = \mathbb{E}_{\substack{z^i, z^{j+} \sim p(z^i, z^j) \\ z^j \sim p(z^j)}} \left[ log \frac{expf(z^i, z^{j+})}{\sum_N expf(z^i, z^j)} \right] \qquad (8)$$

$$I_{ub}(Z^i; Z^j) = \mathbb{E}_{z^i, z^{j+} \sim p(z^i, z^j)} \left[ f^*(z^i, z^{j+}) \right] - \mathbb{E}_{\substack{z^i \sim p(z^i) \\ z^j \sim p(z^j)}} \left[ f^*(z^i, z^j) \right] \qquad (9)$$

where $f(\cdot, \cdot)$ is a score critic approximated by a neural network and $f^*(\cdot, \cdot)$ is the optimal critic from $I_{lb}$ plugged into the $I_{ub}$ objective. $p(z^i, z^j)$ denotes the joint distribution of node representations from views $i$ and $j$, while $p(z^i)$ denotes the marginal distribution. $z^i$ and $z^{j+}$ are mutually positive samples, representing the representations of the same node in views $i$ and $j$ respectively.

To avoid too many extra parameters, the function $f(z^i, z^j)$ is implemented using non-linear projection and cosine similarity. Each term in the total loss $\mathcal{L}$ maximizes mutual information, so we use the lower bound estimator for the calculation. In contrast, we use the upper bound estimator for the generator loss $\mathcal{L}_{gen}$ in InfoMGF-LA, which minimizes mutual information. These two losses can

be expressed as follows:

$$\mathcal{L} = -\frac{2}{V(V-1)} \sum_{i=1}^{V} \sum_{j=i+1}^{V} I_{lb}(Z^i; Z^j) - \frac{1}{V} \sum_{i=1}^{V} I_{lb}(Z^i; Z^{i'}) - \frac{1}{V} \sum_{i=1}^{V} I_{lb}(Z; Z^i) \qquad (10)$$

$$\mathcal{L}_{gen} = \frac{1}{NV} \sum_{i=1}^{V} \sum_{j=1}^{N} \left( 1 - \frac{(X_j^i)^\top \hat{X}_j^i}{\|X_j^i\| \cdot \|\hat{X}_j^i\|} \right) + \lambda * \frac{1}{V} \sum_{i=1}^{V} I_{ub}(Z^i; Z^{i'}) \qquad (11)$$

Finally, we provide the InfoMGF-LA algorithm in Appendix C.1. In Step 1 of each epoch, we keep the augmented graph fixed and optimize both the refined graphs and the fused graph using the total loss $\mathcal{L}$, updating the parameters of Graph Learners and GCN. In Step 2, we keep the refined graphs fixed and optimize each augmented graph using $\mathcal{L}_{gen}$, updating the parameters of the Augmented Graph Generator and Decoder. After training, $G^s$ and $Z$ are used for downstream tasks.

## 4 Experiments

In this section, our aim is to answer three research questions: **RQ1:** How effective is InfoMGF for different downstream tasks in unsupervised settings? **RQ2:** Does InfoMGF outperform baselines of various types under different adversarial attacks? **RQ3:** How do the main modules influence the performance of InfoMGF?

### 4.1 Experimental Setups

**Downstream tasks.** We evaluate the learned graph on node clustering and node classification tasks. For node clustering, following [8], we apply the K-means algorithm on the node representations $Z$ of $G^s$ and use the following four metrics: Accuracy (ACC), Normalized Mutual Information (NMI), F1 Score (F1), and Adjusted Rand Index (ARI). For node classification, following the graph structure learning settings in [16], we train a new GCN on $G^s$ for evaluation and use the following two metrics: Macro-F1 and Micro-F1.

**Datasets.** We conduct experiments on four real-world benchmark multiplex graph datasets, which consist of two citation networks (i.e., ACM [18] and DBLP [18]), one review network Yelp [35] and a large-scale citation network MAG [36]. Details of datasets are shown in Appendix E.1.

**Baselines.** For node clustering, we compare InfoMGF with two single-graph methods (i.e., VGAE [25] and DGI [37]) and seven multiplex graph methods (i.e., O2MAC [26], MvAGC [38], MCGC [39], HDMI [8], MGDCR [9], DMG [3], and BTGF [10]). All the baselines are unsupervised clustering methods. For a fair comparison, we conduct single-graph methods separately for each graph and present the best results.

For node classification, we compare InfoMGF with baselines of various types: three supervised structure-fixed GNNs (i.e., GCN [22], GAT [40] and HAN [41]), six supervised GSL methods (i.e., LDS [42], GRCN [43], IDGL [44], ProGNN [11], GEN [45] and NodeFormer [46]), three unsupervised GSL methods (i.e., SUBLIME [20], STABLE [47] and GSR [48]), and three structure-fixed UMGL methods (i.e., HDMI [8], DMG [3] and BTGF [10]). GCN, GAT, and all GSL methods are single-graph approaches. For unsupervised GSL methods, following [20], we train a new GCN on the learned graph for node classification. For UMGL methods, following [8], we train a linear classifier on the learned representations. Implementation details can be found in Appendix E.2.

### 4.2 Effectiveness Analysis (RQ1)

Table 1 presents the results of node clustering. Firstly, multiplex graph clustering methods outperform single graph methods overall, demonstrating the advantages of leveraging information from multiple sources. Secondly, compared to other multiplex graph methods, both versions of our approach surpass existing state-of-the-art methods. This underscores the efficacy of our proposed graph structure learning, which eliminates task-irrelevant noise and extracts task-relevant information from all graphs, to serve downstream tasks better. Finally, InfoMGF-LA achieves notably superior results, owing to the exceptional capability of the learnable generative graph augmentation in capturing view-unique task-relevant information.

Table 1: Quantitative results (%) on node clustering. The top 3 highest results are highlighted with **red boldface**, red color and **boldface**, respectively. The symbol "OOM" means out of memory.

| Method | ACM NMI | ARI | ACC | F1 | DBLP NMI | ARI | ACC | F1 | Yelp NMI | ARI | ACC | F1 | MAG NMI | ARI | ACC | F1 |
|---|---|---|---|---|---|---|---|---|---|---|---|---|---|---|---|---|
| VGAE | 45.83 | 41.36 | 67.93 | 68.62 | 61.79 | 65.56 | 84.48 | 83.67 | 39.19 | 42.57 | 65.07 | 56.74 | | OOM | | |
| DGI | 52.94 | 47.55 | 65.36 | 57.34 | 65.59 | 70.35 | 86.88 | 86.02 | 39.42 | 42.62 | 65.29 | 56.79 | 53.56 | 42.6 | 59.89 | 57.17 |
| O2MAC | 42.36 | 46.04 | 77.92 | 78.01 | 58.64 | 60.01 | 83.29 | 82.88 | 39.02 | 42.53 | 65.07 | 56.74 | | OOM | | |
| MvAGC | 64.49 | 66.81 | 87.17 | 87.21 | 50.39 | 51.21 | 78.39 | 77.84 | 24.39 | 29.25 | 63.14 | 56.7 | | OOM | | |
| MCGC | 60.21 | 50.72 | 65.62 | 54.78 | 65.56 | 71.51 | 87.96 | 87.47 | 38.35 | 35.17 | 65.61 | 57.49 | | OOM | | |
| HDMI | 65.44 | 68.87 | 88.11 | 88.14 | 64.85 | 70.85 | 87.39 | 86.75 | 60.81 | 59.35 | 79.56 | 77.6 | 48.15 | 34.92 | 51.78 | 49.8 |
| MGDCR | 58.8 | 55.15 | 73.82 | 70.34 | 62.47 | 62.22 | 81.91 | 80.16 | 44.23 | 46.47 | 72.71 | 54.43 | 54.43 | 43.98 | 61.37 | 60.53 |
| DMG | 64.14 | 67.21 | 87.11 | 87.23 | 69.03 | 73.07 | 88.45 | 87.88 | 65.66 | 66.33 | 88.26 | 89.27 | 48.72 | 39.77 | 61.61 | 60.16 |
| BTGF | 68.92 | 73.14 | 90.09 | 90.11 | 66.28 | 72.47 | 88.05 | 87.28 | 69.97 | 73.53 | 91.39 | 92.32 | | OOM | | |
| InfoMGF-RA | 74.89 | 81.09 | 92.82 | 92.89 | 70.19 | 73.49 | 88.72 | 88.31 | 72.67 | 74.66 | 91.85 | 92.86 | 56.65 | 45.25 | 64.13 | 63.09 |
| InfoMGF-LA | 76.53 | 81.49 | 93.45 | 93.42 | 73.22 | 78.49 | 91.08 | 90.69 | 75.18 | 78.91 | 93.26 | 94.01 | | OOM | | |

Table 2: Quantitative results with standard deviation ($\% \pm \sigma$) on node classification. Available data for GSL during training is shown in the first column, supervised methods depend on Y for GSL. The symbol "-" indicates that the method is structure-fixed, which does not learn a new structure.

| Available Data for GSL | Methods | ACM Macro-F1 | Micro-F1 | DBLP Macro-F1 | Micro-F1 | Yelp Macro-F1 | Micro-F1 | MAG Macro-F1 | Micro-F1 |
|---|---|---|---|---|---|---|---|---|---|
| - | GCN | 90.27±0.59 | 90.18±0.61 | 90.01±0.32 | 90.99±0.28 | 78.01±1.89 | 81.03±1.81 | 75.98±0.07 | 75.76±0.10 |
| - | GAT | 91.52±0.62 | 91.46±0.62 | 90.22±0.37 | 91.13±0.40 | 82.12±1.47 | 84.43±1.56 | OOM | |
| - | HAN | 91.67±0.39 | 91.47±0.62 | 90.53±0.24 | 91.47±0.22 | 88.49±1.73 | 88.78±1.40 | OOM | |
| X,Y,A | LDS | 92.35±0.43 | 92.05±0.26 | 88.11±0.86 | 88.74±0.85 | 75.98±2.35 | 78.14±1.98 | OOM | |
| X,Y,A | GRCN | 93.04±0.17 | 92.94±0.18 | 88.33±0.47 | 89.43±0.44 | 76.05±1.05 | 80.68±0.96 | OOM | |
| X,Y,A | IDGL | 91.69±1.24 | 91.63±1.24 | 89.65±0.60 | 90.61±0.56 | 76.98±5.78 | 79.15±5.06 | OOM | |
| X,Y,A | ProGNN | 90.57±1.03 | 90.50±1.29 | 83.13±1.56 | 84.83±1.36 | 51.76±1.46 | 58.39±1.25 | OOM | |
| X,Y,A | GEN | 87.91±2.78 | 87.88±2.61 | 89.74±0.69 | 90.65±0.71 | 80.43±3.78 | 82.68±2.84 | OOM | |
| X,Y,A | NodeFormer | 91.33±0.77 | 90.60±0.95 | 79.54±0.78 | 80.56±0.62 | 91.69±0.65 | 90.59±1.21 | 77.21±0.18 | 77.08±0.19 |
| X,A | SUBLIME | 92.42±0.16 | 92.13±0.37 | 90.98±0.37 | 91.82±0.27 | 79.68±0.79 | 82.99±0.82 | 75.96±0.05 | 75.71±0.03 |
| X,A | STABLE | 83.54±4.20 | 83.38±4.51 | 75.18±1.95 | 76.42±1.95 | 71.48±4.71 | 76.62±2.75 | OOM | |
| X,A | GSR | 92.14±1.08 | 92.11±0.99 | 76.59±0.45 | 77.69±0.42 | 83.85±0.76 | 85.73±0.54 | OOM | |
| - | HDMI | 91.01±0.32 | 90.86±0.31 | 89.91±0.49 | 90.89±0.51 | 80.73±0.64 | 84.05±0.91 | 72.22±0.14 | 71.84±0.15 |
| - | DMG | 90.42±0.36 | 90.31±0.35 | 90.42±0.57 | 91.34±0.49 | 91.61±0.62 | 90.24±0.81 | 76.34±0.09 | 76.13±0.10 |
| - | BTGF | 91.75±0.11 | 91.62±0.11 | 90.71±0.24 | 91.57±0.21 | 92.81±1.12 | 91.37±1.28 | OOM | |
| X,A | InfoMGF-RA | 93.21±0.22 | 93.14±0.21 | 90.99±0.36 | 91.93±0.29 | 93.09±0.27 | 92.02±0.34 | 77.25±0.06 | 77.11±0.06 |
| X,A | InfoMGF-LA | 93.42±0.21 | 93.35±0.21 | 91.28±0.31 | 92.12±0.28 | 93.26±0.26 | 92.24±0.34 | OOM | |

Table 2 reports the node classification results. Overall, GSL methods outperform structure-fixed methods, demonstrating the unreliability of the original structure in real-world data and the significance of graph structure learning. Particularly for various carefully designed UMGL methods, the original graphs with rich task-irrelevant noise severely limit their performance. Compared to existing single-graph GSL methods, both versions of InfoMGF outperform the supervised methods. By capturing shared and unique information from multiplex graphs, InfoMGF can integrate more comprehensive task-relevant information. Finally, we can observe that the proposed InfoMGF-LA with learnable augmentation indeed surpasses the random augmentation version, once again highlighting its advantage in exploring task-relevant information.

We select a subgraph from the ACM dataset with nodes in two classes (database (C1) and data mining (C2)) and visualize the edge weights in the original multiplex graphs and the fused graph learned by InfoMGF-LA. From Figure 3, the learned graph mainly consists of intra-class edges. Compared

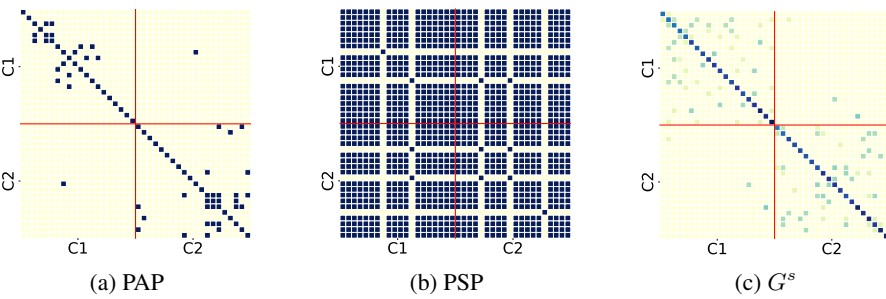

(a) PAP      (b) PSP      (c) $G^{rs}$

Figure 3: Heatmaps of the subgraph adjacency matrices of the original and learned graphs on ACM.

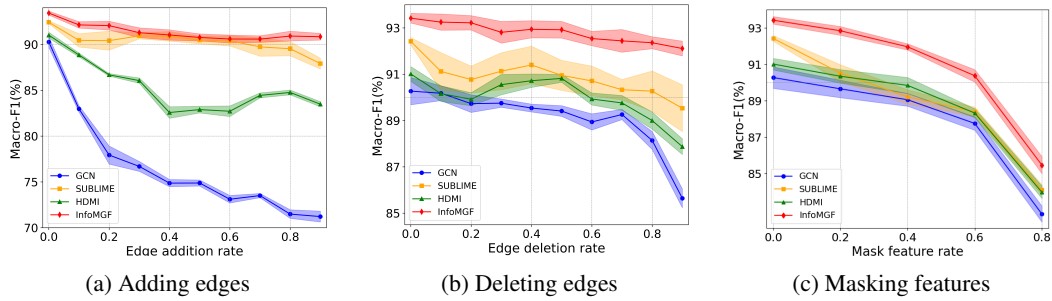

(a) Adding edges      (b) Deleting edges      (c) Masking features

Figure 4: Robustness analysis on ACM.

to the nearly fully connected PSP view, InfoMGF significantly reduces inter-class edges, reflecting our effective removal of task-irrelevant noise. Compared to the PAP view, InfoMGF introduces more intra-class edges, benefiting from capturing shared and unique task-relevant information from all graphs. Furthermore, varying edge weights in $G^s$ represent different importance levels, better serving downstream tasks. In summary, the above experiment results across various downstream tasks demonstrate the effectiveness of InfoMGF. We use the InfoMGF-LA version in the subsequent sections to conduct more comprehensive analyses.

### 4.3 Robustness Analysis (RQ2)

To evaluate the robustness of InfoMGF against random noise, we perturb each graph on the ACM dataset by randomly adding edges, deleting edges, and masking features. We compare InfoMGF against various baselines: structure-fixed method (GCN), GSL method (SUBLIME), and UMGL method (HDMI). From Figure 4a and 4b, it is evident that with increasing rates of edge perturbing, the performance of each method deteriorates, while the GSL methods (i.e., InfoMGF and SUB-LIME) exhibit better robustness. Notably, InfoMGF **consistently outperforms all other methods** across both experimental settings, especially when the perturbation rate is extremely high.

Figure 4c shows the performance of InfoMGF and various baselines when injecting random feature noise. It can be observed that InfoMGF exhibits excellent robustness against feature noise, while the performance of SUBLIME degrades rapidly. As a single graph structure learning method, SUB-LIME's performance heavily relies on the quality of node features. In contrast, our method can directly optimize task-relevant information in multi-view graph structures (e.g., edges shared across multiple graphs are likely to share task-relevant information, which can be directly learned through $\mathcal{L}_s$), thus reducing dependence on node features. Consequently, InfoMGF demonstrates superior robustness against various types of noise.

### 4.4 Ablation Study (RQ3)

Table 3: Performance ($\% \pm \sigma$) of InfoMGF and its variants.

| Variants | ACM | | DBLP | | Yelp | |
|---|---|---|---|---|---|---|
| | Macro-F1 | Micro-F1 | Macro-F1 | Micro-F1 | Macro-F1 | Micro-F1 |
| w/o $\mathcal{L}_s$ | 93.05±0.49 | 92.98±0.49 | 90.44±0.45 | 91.39±0.41 | 93.15±0.12 | 92.11±0.13 |
| w/o $\mathcal{L}_u$ | 92.66±0.53 | 92.61±0.51 | 90.13±0.43 | 91.05±0.44 | 92.23±0.27 | 90.96±0.36 |
| w/o Aug. | 92.84±0.17 | 92.81±0.16 | 90.94±0.45 | 91.81±0.41 | 92.76±0.49 | 91.63±0.51 |
| w/o Rec. | 92.91±0.53 | 92.88±0.51 | 91.05±0.27 | 91.87±0.23 | 92.65±0.27 | 91.45±0.37 |
| InfoMGF | 93.42±0.21 | 93.35±0.21 | 91.28±0.31 | 92.12±0.28 | 93.26±0.26 | 92.24±0.34 |

To verify the effectiveness of each part of InfoMGF, we design four variants and compare the classification performance against InfoMGF.

*Effectiveness of loss components.* Recall InfoMGF maximizes view-shared and unique task-relevant information by $\mathcal{L}_s$ and $\mathcal{L}_u$. Thus, we design two variants (w/o $\mathcal{L}_s$ and w/o $\mathcal{L}_u$). Table 3 shows the necessity of each component. Furthermore, we can observe that the removal of $\mathcal{L}_u$ has a greater impact compared to $\mathcal{L}_s$, which can be explained by the fact that optimization of $\mathcal{L}_u$ actually maximizes the overall task-relevant information of each view, rather than the unique aspects of the view.

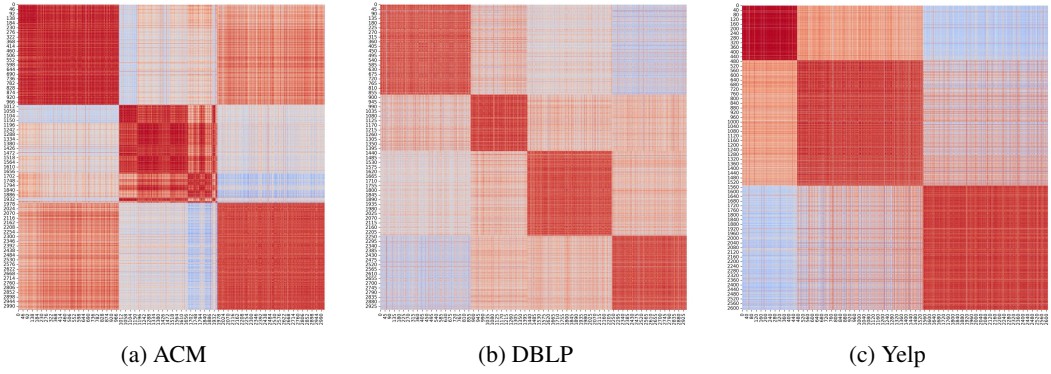

|           |           |          |
|-----------|-----------|----------|
| (a) ACM   | (b) DBLP  | (c) Yelp |

Figure 5: Node correlation maps of representations reordered by node labels.

*Effectiveness of augmentation module.* The InfoMGF-LA framework incorporates learnable generative augmentation and maximizes the mutual information $I(G_i^s; G_i')$ to mine the task-relevant information. We first compare InfoMGF with maximizing the mutual information $I(G_i^s; G_i)$ with the original graph structure without augmentation (w/o Aug.). Furthermore, we remove the reconstruction loss term (w/o Rec.) of $\mathcal{L}_{gen}$ to analyze the necessity of preserving crucial information. The results show that maximizing $I(G_i^s; G_i)$ leads to poorer performance compared to $I(G_i^s; G_i')$, consistent with Theorem 2. Meanwhile, deleting the reconstruction term from $\mathcal{L}_{gen}$ also results in the augmented graph lacking task-relevant information, thus hurting model performance.

## 4.5 Node Correlation Visualization

We further visualize the node correlation in the learned representations $Z$ of the fused graph, which is used in the clustering task. Figure 5 shows the node correlation heatmaps of the representations, where both rows and columns are reordered by the node labels. In the heatmap, warmer colors signify a higher correlation between nodes. It is evident that the correlation among nodes of the same class is significantly higher than that of nodes from different classes. This is due to $G^s$ mainly containing intra-class edges without irrelevant inter-class edges, which validates the effectiveness of InfoMGF in unsupervised graph structure learning.

## 5 Conclusion and Limitation

This paper delves into the unsupervised graph structure learning within multiplex graphs for the first time. The proposed InfoMGF refines the graph structure to eliminate task-irrelevant noise, while simultaneously maximizing both the shared and unique task-relevant information across different graphs. The fused graph applied to downstream tasks is optimized to incorporate clean and comprehensive task-relevant information from all graphs. Theoretical analyses and extensive experiments ensure the effectiveness of InfoMGF. A limitation of our research lies in its focus solely on the pure unsupervised scenario. In some real-world scenarios where partial node labels are available, label information can be used to learn a better structure of multiplex graphs. Such supervised or semi-supervised problems are left for future exploration.

## Acknowledgments and Disclosure of Funding

This work was supported by the National Natural Science Foundation of China (No. 62276053).

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

# A  Notations

Table 4: Frequently used notations.

| Notation | Description |
|---|---|
| $G_v = \{A_v, X\}$ | The $v$-th original graph. |
| $Y$ | The label information. |
| $V, N, d_f$ | The number of graphs/nodes/features. |
| $A_v \in \{0,1\}^{N \times N}$ | The adjacency matrix of $v$-th original graph. |
| $X \in \mathbb{R}^{N \times d_f}$ | The shared feature matrix across all graphs. |
| $G'_v = \{A'_v, X'\}$ | The $v$-th augmented graph. |
| $G^s_v = \{A^s_v, X\}$ | The $v$-th refined graph. |
| $G^s = \{A_s, X\}$ | The learned fused graph. |
| $H^v \in \mathbb{R}^{N \times d_f}$ | The node embeddings of the original graph from the graph learner. |
| $Z^v \in \mathbb{R}^{N \times d}$ | The node representations of the refined graph of the GCN encoder. |
| $Z \in \mathbb{R}^{N \times d}$ | The node representations of the fused graph from the GCN encoder. |
| $\vec{m} \in \{0,1\}^{d_f}$ | The random masking vector for feature masking. |
| $M \in \{0,1\}^{N \times N}$ | The random masking matrix for edge dropping. |
| $r$ | The order of graph aggregation in SGC. |
| $L$ | The number of layers in GCN. |
| $k$ | The number of neighbors in $k$NN. |
| $\lambda$ | The positive hyper-parameter in $\mathcal{L}_{gen}$. |
| $I(G^s_i; G^s_j)$ | The mutual information between the $i$-th and $j$-th refined graphs. |
| $\mathcal{L}$ | The total loss of InfoMGF-RA and InfoMGF-LA. |
| $\mathcal{L}_{gen}$ | The loss of augmented graph generator in InfoMGF-LA. |
| $\odot$ | The Hadamard product. |
| $\sigma(\cdot)$ | The non-linear activation function. |
| $Bern(\cdot)$ | The Bernoulli distribution. |
| $[\cdot; \cdot]$ | The concatenation operation. |

# B  Related Work

**Unsupervised Multiplex Graph Learning (UMGL).** Unlike supervised methods such as HAN [41] and SSAMN [49] which rely on label information, UMGL tackles unsupervised tasks in multiplex graphs by using node features and graph structures [50]. Early UMGL methods such as MvAGC [38] and MCGC [39] combine graph filtering with unsupervised techniques such as spectral and subspace clustering to uncover underlying patterns in complex networks. With the rise of deep representation learning [51], UMGL has embraced a new paradigm: Unsupervised learning of low-dimensional node representations using graph neural networks (GNN) [4] and self-supervised techniques [5] for downstream tasks such as node classification, node clustering, and similarity search. O2MAC [26] pioneered the use of GNNs in UMGL, selecting the most informative graph and reconstructing all graph structures to capture shared information. DMGI [2] and HDMI [8] maximize mutual information between local and global contexts, then fuse representations from different relations. MGCCN [52], MGDCR [9], and BTGF [10] employ various contrastive losses to align representations of diverse relations and prevent dimension collapse. CoCoMG [53] and DMG [3] capture complete information by learning consistency and complementarity between graphs. Despite these advances, a critical factor that limits the performance of UMGL is overlooked: the reliability of graph structures, which is the focus of our research.

**Graph Structure Learning (GSL).** With the advancement of graph neural networks, instead of designing complex neural architectures as model-centric approaches, some data-centric research has focused on the graph data itself [54], with graph structure learning (GSL) gaining widespread attention for studying the reliability of graph structures. GSL, based on empirical analysis of graph data, recognizes that real-world graph structures are often unreliable, thus opting to learn new structures. GSLB [16] summarizes the general framework of graph structure learning: a Graph Learner takes in the original graph $G = \{A, X\}$ and generates a refined graph $G^s = \{A^s, X\}$; then, a Graph Encoder uses the refined graph to obtain node representations or perform class prediction. Consequently, GSL can be broadly categorized into supervised and unsupervised methods based on whether label information is utilized to learn the new structure. For supervised GSL, probabilistic models like LDS [42] and GEN [45] are employed to generate graph structures; GRCN [43], IDGL [44], and NodeFormer [46] calculate node similarities through metric learning or scalable attention

mechanisms; while ProGNN [11] directly treats all elements in the adjacency matrix as learnable parameters. Meanwhile, methods like SUBLIME [20], STABLE [47], and GSR [48] introduce self-supervised signals through contrastive learning to learn graph structures without requiring label information. Almost all existing GSL studies concentrate on a single homogeneous graph, with only a handful of works such as GTN [18] and HGSL [55] attempting supervised structure learning on heterogeneous graphs containing multiple types of nodes. There is still a lack of research concerning more practically significant unsupervised graph structure learning within multiplex graphs.

**Contrastive Learning and Information Theory.** Contrastive learning, as an effective paradigm of self-supervised learning, enables representation learning without labeled information [56]. It has found widespread applications across various modalities [57–59], particularly effective in multi-view or multi-modal tasks [60–62]. Its theoretical foundation is rooted in multi-view information theory [63, 30, 64]. Standard contrastive learning is based on the assumption of multi-view redundancy: shared information between views is almost exactly what is relevant for downstream tasks [17, 23, 24, 65]. They capture shared task-relevant information between views through contrastive pre-training, thus achieving data compression and sufficient representation learning. To successfully apply contrastive learning to multi-modal data with task-relevant unique information, some studies have improved the framework of contrastive learning and extended it to multi-view non-redundancy [17, 28]. Recent efforts also attempt to apply contrastive learning to graph learning tasks [66]. They generate contrastive views through graph data augmentation [67] or directly utilize different relations within graph data [39]. However, existing multi-view graph contrastive learning still suffers from the limitation of multi-view redundancy, failing to extract view-unique task-relevant information effectively.

# C Algorithm and Methodology Details

## C.1 Algorithm

---

**Algorithm 1:** The optimization of InfoMGF-RA

---

**Input:** Original graph structure $G = \{G_1, ..., G_V\}$; Number of nearest neighbors $k$; Random masking probability $\rho$; Number of epochs $E$

**Output:** Learned fused graph $G^s$ and node representations $Z$

1   Initialize parameters;
2   Obtain view-specific node features $\{X^1, \cdots, X^V\}$ by Eq.(1);
3   **for** $e = 1, 2, 3, ..., E$ **do**
4     **for** *each view $v$ in* $\{1, \cdots, V\}$ **do**
5       Generate refined graph $G_v^{'s} = \{A_v^s, X\}$ with graph learner by Eq.(1) and post-processors;
6       Generate augmented graph $G_v' = \{A_v', X'\}$ with random feature masking and edge dropping;
7     **end**
8     Generate fused graph $G^s = \{A^s, X\}$ with graph learner by Eq.(6) and post-processors;
9     Obtain node representations $\{Z^1, \cdots, Z^V, Z^{1'}, \cdots, Z^{V'}, Z\}$ through graph encoder GCN;
10    Calculate the total loss $\mathcal{L}$ by Eq.(10) and update parameters in GCN and graph learners;
11 **end**
12 return fused graph $G^s$ and node representations $Z$;

---

**Algorithm 2:** The optimization of InfoMGF-LA

---

**Input:** Original graph structure $G = \{G_1, ..., G_V\}$; Number of nearest neighbors $k$; Feature masking probability $\rho$; Hyper-parameter $\lambda$; Number of epochs $E$

**Output:** Learned fused graph $G^s$ and node representations $Z$

1   Initialize parameters;

2   Obtain view-specific node features $\{X^1, \cdots, X^V\}$ by Eq.(1);

3   **for** $e = 1, 2, 3, ..., E$ **do**

      // Step 1: Fix augmented graphs $\{G'_1, \cdots, G'_V\}$

4      **for** *each view $v$ in* $\{1, \cdots, V\}$ **do**

5         Generate refined graph $G^s_v = \{A^s_v, X\}$ with graph learner by Eq.(1) and post-processors;

6      **end**

7      Generate fused graph $G^s = \{A^s, X\}$ with graph learner by Eq.(6) and post-processors;

8      Obtain node representations $\{Z^1, \cdots, Z^V, Z^{1'}, \cdots, Z^{V'}, Z\}$ through graph encoder GCN;

9      Calculate the total loss $\mathcal{L}$ by Eq.(10) and update parameters in GCN and graph learners;

      // Step 2: Fix refined graphs and fused graph $\{G^s_1, \cdots, G^s_V, G^s\}$

10     **for** *each view $v$ in* $\{1, \cdots, V\}$ **do**

11       Generate augmented graph $G'_v = \{A'_v, X'\}$ with random feature masking and augmented graph generator in Section 3.3

12     **end**

13     Obtain node representations $\{Z^1, \cdots, Z^V, Z^{1'}, \cdots, Z^{V'}\}$ through graph encoder GCN;

14     Obtain reconstructed features $\{\hat{X}^1, \cdots, \hat{X}^V\}$ through decoder;

15     Calculate $\mathcal{L}_{gen}$ by Eq.(11) and update parameters in augmented graph generator and decoder;

16   **end**

17   return fused graph $G^s$ and node representations $Z$;

---

### C.2   Complexity Analysis

First, we analyze the time complexity of each component in InfoMGF. In this paragraph, let $V$, $N$, and $m$ represent the numbers of graphs, nodes, and edges, while $b_1$ and $b_2$ denote the batch sizes of the locality-sensitive $k$ NN and contrastive loss computation. The layer numbers of graph learner, graph encoder GCN, and non-linear projector are denoted as $L_1$, $L_2$, and $L_3$, respectively. The feature, hidden layer, and representation dimensions are denoted as $d_f$, $d_h$, and $d$, respectively. We analyze the complexity of $k$NN and GCN in scalable versions. Before training, scalable SGC is applied with a complexity of $\mathcal{O}(Vmrd_f)$ related to the aggregation order $r$. During training, we first perform a graph learner with scalable $k$ NN that requires $\mathcal{O}(VNL_1d_f + VNb_1d_f)$. For the GCN encoder and non-linear projector, the total complexity is $\mathcal{O}\left(VmL_2d_h + Vmd + VNL_2d_h^2 + VNd_h(d + d_f) + VNL_3d^2\right)$. Within the graph augmentation module, the complexity of feature masking is $\mathcal{O}(Nd_f)$. The learnable generative graph augmentation in InfoMGF-LA has a complexity of $\mathcal{O}(VNd_fd_h + Vmd_h + VNd_fd)$, where the first two terms are contributed by the augmented graph generator and the last one is for the decoder. For InfoMGF-RA, the random edge drop requires $\mathcal{O}(Vm)$ time complexity. For the loss computation, the complexity is $\mathcal{O}(V^2Nb_2d)$.

To simplify the overall complexity, we denote the larger terms within $L_1$, $L_2$, and $L_3$ as $L$, the larger terms between $d_h$ and $d$ as $\hat{d}$, the larger terms between $b_1$ and $b_2$ as $B$. Since the scalable SGC operation only needs to be performed once before training, its impact on training time is negligible. Therefore, we only consider total complexity during the training process. The overall complexity of both InfoMGF-RA and InfoMGF-LA is $\mathcal{O}(VmL\hat{d} + VNL\hat{d}^2 + VNd_f(\hat{d} + L) + VNB(d_f + V\hat{d}))$, which is comparable to the mainstream unsupervised GSL models, including our baselines. For example, SUBLIME [20] needs to be trained on each graph in a multiplex graph dataset, and its time complexity is $\mathcal{O}(VmL\hat{d} + VNL\hat{d}^2 + VNd_f(\hat{d} + L) + VNB(d_f + \hat{d}))$, which only has a slight difference in the last term compared to the time complexity of our method.

## C.3 Details of Post-processing Techniques

After constructing the cosine similarity matrix of $H^v$, we employ the postprocessor to ensure that $A_v^s$ is sparse, nonnegative, symmetric and normalized. For convenience, we omit the subscript $v$ in the discussion below.

$k$**NN for sparsity.** The fully connected adjacency matrix usually makes little sense for most applications and results in expensive computation cost. Hence, we conduct the $k$-nearest neighbors ($k$NN) operation to sparsify the learned graph. We keep the edges with top-$k$ values and otherwise to $0$ for each node and get the sparse adjacency matrix $A^{sp}$.

**Symmetrization and Activation.** As real-world connections are often bidirectional, we make the adjacency matrix symmetric. Additionally, the weight of each edge should be non-negative. With the input $A^{sp}$, they can be expressed as follows:

$$A^{sym} = \frac{\sigma(A^{sp}) + \sigma(A^{sp})^\top}{2} \tag{12}$$

where $\sigma(\cdot)$ is a non-linear activation implemented by the ReLU function.

**Normalization.** The normalized adjacency matrix with self-loop can be obtained as follows:

$$A^s = (\tilde{D}^{sym})^{-\frac{1}{2}} \tilde{A}^{sym} (\tilde{D}^{sym})^{-\frac{1}{2}} \tag{13}$$

where $\tilde{D}^{sym}$ is the degree matrix of $\tilde{A}^{sym}$ with self-loop. Afterward, we can obtain the adjacency matrix $A_v^s$ for each view, which possesses the desirable properties of sparsity, non-negativity, symmetry, and normalization.

## C.4 Details of Loss Functions

For each view $i$ and $j$, the lower and upper bound of $I(Z^i; Z^j)$ in Eq.(8) and Eq.(9) can be calculated for the node $m$:

$$\ell_{lb}(Z_m^i, Z_m^j) = log \frac{e^{sim(\tilde{Z}_m^i, \tilde{Z}_m^j)/\tau_c}}{\sum_{n=1}^{N} e^{sim(\tilde{Z}_m^i, \tilde{Z}_n^j)/\tau_c}} \tag{14}$$

$$\ell_{ub}(Z_m^i, Z_m^j) = sim(\tilde{Z}_m^i, \tilde{Z}_m^j)/\tau_c - \frac{1}{N} \sum_{n=1}^{N} sim(\tilde{Z}_m^i, \tilde{Z}_n^j)/\tau_c, \tag{15}$$

where $\tilde{Z}_m^i$ is the non-linear projection of $Z_m^i$ through MLP, $sim(\cdot)$ refers to the cosine similarity and $\tau_c$ is the temperature parameter in contrastive loss. The loss $\mathcal{L}_s$ is computed as follows:

$$\mathcal{L}_s = -\frac{1}{NV(V-1)} \sum_{i=1}^{V} \sum_{j=i+1}^{V} \sum_{m=1}^{N} (\ell_{lb}(Z_m^i, Z_m^j) + \ell_{lb}(Z_m^j, Z_m^i)). \tag{16}$$

Likewise, we can compute $\mathcal{L}_f$ and $\mathcal{L}_u$ in the total loss $\mathcal{L}$ with the same approach. Upon optimizing $\mathcal{L}$, our objective also entails the minimization of $\mathcal{L}_{gen}$, which incorporates $\lambda * \mathcal{L}_u$ (here we compute $\mathcal{L}_u$ using the upper bound) and the loss term of the reconstruction. $\mathcal{L}_{gen}$ can be represented by:

$$\mathcal{L}_{gen} = \lambda * \frac{1}{2NV} \sum_{i=1}^{V} \sum_{j=1}^{N} (\ell_{ub}(Z_j^i, Z_j^{i'}) + \ell_{ub}(Z_j^{i'}, Z_j^i)) + \frac{1}{NV} \sum_{i=1}^{V} \sum_{j=1}^{N} \left( 1 - \frac{(X_j^i)^\top \hat{X}_j^i}{\|X_j^i\| \cdot \|\hat{X}_j^i\|} \right) \tag{17}$$

## D Proofs of Theorems

### D.1 Properties of multi-view mutual information and representations

In this section, we enumerate some basic properties of mutual information used to prove the theorems. For any random variables $x, y$ and $z$, we have:

$(P_1)$ Non-negativity:

$$I(x; y) \geq 0, I(x; y|z) \geq 0 \tag{18}$$

($P_2$) Chain rule:

$$I(x, y; z) = I(y; z) + I(x; z|y) \tag{19}$$

($P_3$) Chain rule (Multivariate Mutual Information):

$$I(x; y; z) = I(y; z) - I(y; z|x) \tag{20}$$

We also introduce the property of representation:

**Lemma 1.** *[63, 68] If $z$ is a representation of $v$, then:*

$$I(z; a|v, b) = 0 \tag{21}$$

*for any variable (or groups of variables) $a$ and $b$ in the system. Whenever a random variable $z$ is defined as a representation of $v$, we state that $z$ is conditionally independent of any other variable in the system given $v$. This does not imply that $z$ must be a deterministic function of $v$, but rather that the source of $z$'s stochasticity is independent of the other random variables.*

### D.2   Proof of Proposition 1

*Proof of Proposition 1:*  Due to each $G_i^s$ is obtained from $G_i$ through a deterministic function, which is independent of other variables. Thus, here $G_i^s$ can be regarded as a representation of $G_i$. For any two different views $G_i$ and $G_j$, we have:

$$
\begin{aligned}
I(G_i^s; G_j) &\stackrel{(P_2)}{=} I(G_i^s; G_j^s, G_j) - I(G_i^s; G_j^s|G_j) \\
&=^* I(G_i^s; G_j^s, G_j) \\
&= I(G_i^s; G_j^s) + I(G_i^s; G_j|G_j^s) \\
&\geq I(G_i^s; G_j^s)
\end{aligned}
\tag{22}
$$

where $*$ follows from Lemma 1. The bound reported in this equation is tight when $I(G_i^s; G_j|G_j^s) = 0$, this happens whenever $G_j^s$ contains all the information regarding $G_i^s$ (and therefore $G_i$). Symmetrically, we can also prove $I(G_j^s; G_i) \geq I(G_i^s; G_j^s)$, then we have

$$I(G_i^s; G_j) + I(G_j^s; G_i) \geq 2I(G_i^s; G_j^s) \tag{23}$$

Proposition 1 holds.

### D.3   Proof of Theorem 1

*Proof of Theorem 1.*  From the definition of optimal augmentation graph, we have

$$I(G_i'; G_i) = I(Y; G_i) \tag{24}$$

Similar to the proof of Proposition 1, as $G_i^s$ is regarded as a representation of $G_i$, therefore:

$$I(G_i^s; Y|G_i) = 0 \tag{25}$$

$$I(G_i^s; G_i'|G_i) = 0 \tag{26}$$

Based on Eq.(24) and the above two equations, then

$$
\begin{aligned}
I(G_i^s; G_i') &= I(G_i; G_i^s; G_i') + I(G_i^s; G_i'|G_i) \\
&\stackrel{Eq.(26)}{=} I(G_i; G_i') - I(G_i; G_i'|G_i^s) \\
&\stackrel{Eq.(24)}{=} I(G_i; Y) - I(G_i; Y|G_i^s) \\
&\stackrel{(P_3)}{=} I(G_i; Y; G_i^s) \\
&\stackrel{Eq.(25)}{=} I(G_i; Y; G_i^s) + I(G_i^s; Y|G_i) \\
&\stackrel{(P_3)}{=} I(G_i^s; Y)
\end{aligned}
\tag{27}
$$

It shows that maximizing $I(G_i^s; G_i')$ and maximizing $I(G_i^s; Y)$ are equivalent. Theorem 1 holds.

## D.4 Proof of Theorem 2

*Proof of Theorem 2.* Here we theoretically compare $I(G_i^s; G_i)$ with $I(G_i^s; G_i')$.

*Discussion 1.* For $I(G_i^s; G_i)$, we have:

$$
\begin{aligned}
I(G_i^s; G_i) &= I(G_i; Y; G_i^s) + I(G_i^s; G_i|Y) \\
&= I(G_i^s; Y) - I(G_i^s; Y|G_i) + I(G_i^s; G_i|Y) \\
&= I(G_i^s; Y) + I(G_i^s; G_i|Y)
\end{aligned}
\tag{28}
$$

In the process of maximizing $I(G_i^s; G_i)$, not only is task-relevant information (the first term) maximized, but task-irrelevant information (the second term) is also maximized.

*Discussion 2.* For $I(G_i^s; G_i')$, based on Theorem 1, we have:

$$
I(G_i^s; G_i') = I(G_i^s; Y)
\tag{29}
$$

Obviously, no task-irrelevant information is maximized. Theorem 2 holds.

## D.5 Proof of Theorem 3

*Proof of Theorem 3.* To prove the theorem, we need to use the following three properties of entropy:

($H_1$) Relationship between the mutual information and entropy:

$$
I(x; y) = H(x) - H(x|y)
\tag{30}
$$

($H_2$) Relationship between the conditional entropy and entropy:

$$
H(x|y) = H(x, y) - H(y)
\tag{31}
$$

($H_3$) Relationship between the conditional mutual information and entropy:

$$
I(x; y|z) = H(x|z) - H(x|y, z)
\tag{32}
$$

By maximizing the mutual information with each refined graph, the optimized fused graph $G^s$ would contain all information from every $G_i^s$. For any $G_i^s$, we denote $G_c^s$ as the fused graph of all views except view $i$. Thus we have:

$$
H(G^s) = H(G_i^s|G_c^s) + H(G_c^s|G_i^s) + I(G_i^s; G_c^s)
\tag{33}
$$

where $H(G_i^s|G_c^s)$ and $H(G_c^s|G_i^s)$ indicate the specific information of $G_c^s$ and $G_i^s$ respectively, and $I(G_i^s; G_c^s)$ indicates the consistent information between $G_c^s$ and $G_i^s$.

Then we have:

$$
\begin{aligned}
H(G^s) &= H(G_i^s|G_c^s) + H(G_c^s|G_i^s) + I(G_i^s; G_c^s) \\
&\overset{(H_1)}{=} H(G_i^s|G_c^s) + H(G_c^s|G_i^s) + H(G_i^s) - H(G_i^s|G_c^s) \\
&\overset{(H_2)}{=} H(G_c^s|G_i^s) + H(G_i^s, G_c^s) - H(G_c^s|G_i^s) \\
&= H(G_i^s, G_c^s)
\end{aligned}
\tag{34}
$$

Therefore, for any downstream task $Y$, we further have:

$$
H(G^s, Y) = H(G_i^s, G_c^s, Y).
\tag{35}
$$

Based on the properties of mutual information and entropy, we can prove:

$$
\begin{aligned}
I(G^s; Y) &= H(G^s) - H(G^s|Y) \\
&= H(G^s) - H(G^s, Y) + H(Y) \\
&\overset{Eq.(34)}{=} H(G_c^s, G_i^s) - H(G_i^s, G_c^s, Y) + H(Y)
\end{aligned}
\tag{36}
$$

Based on the properties of entropy, we have the proofs as follows:

$$
I(G_i^s; Y) = H(G_i^s) - H(G_i^s|Y)
\tag{37}
$$

$$I(G_c^s; Y|G_i^s) = H(G_c^s|G_i^s) - H(G_c^s|G_i^s, Y)$$
$$= H(G_i^s, G_c^s) - H(G_i^s) - H(G_c^s|G_i^s, Y) \tag{38}$$

With the equations above, we can obtain

$$
\begin{aligned}
I(G_i^s; Y) + I(G_c^s; Y|G_i^s) &= H(G_i^s) - H(G_i^s|Y) + H(G_i^s, G_c^s) - H(G_i^s) - H(G_c^s|G_i^s, Y) \\
&= H(G_i^s, G_c^s) - H(G_i^s|Y) - H(G_c^s|G_i^s, Y) \\
&= H(G_i^s, G_c^s) - H(G_i^s, Y) + H(Y) - H(G_c^s|G_i^s, Y) \\
&\overset{(H_2)}{=} H(G_i^s, G_c^s) - H(G_i^s, Y) + H(Y) - H(G_i^s, G_c^s, Y) + H(G_i^s, Y) \\
&= H(G_i^s, G_c^s) + H(Y) - H(G_i^s, G_c^s, Y)
\end{aligned}
\tag{39}
$$

According to Eq.(36) and Eq.(39), we have:

$$I(G^s; Y) = I(G_i^s; Y) + I(G_c^s; Y|G_i^s). \tag{40}$$

As $I(G_c^s; Y|G_i^s) \geq 0$ $(P_1)$, then we can get

$$I(G^s; Y) \geq I(G_i^s; Y). \tag{41}$$

Similarly, we can also obtain

$$I(G^s; Y) \geq I(G_c^s; Y). \tag{42}$$

As Eq.(41) holds for any $i$, thus

$$I(G^s; Y) \geq \max_i I(G_i^s; Y). \tag{43}$$

Theorem 3 holds.

# E  Experimental Settings

## E.1  Datasets

We consider 4 benchmark datasets in total. The statistics of the datasets are provided in Table 5. Through the value of "Unique relevant edge ratio", we can observe a significant amount of view-unique task-relevant information present in each real-world multiplex graph dataset. It should be noted that MAG is a subset of OGBN-MAG [36], consisting of the four largest classes. This dataset was first organized into its current subset version in the following paper [1].

Table 5: Statistics of datasets.

| Dataset | Nodes | Relation type | Edges | Unique relevant edge ratio (%) | Features | Classes | Training | Validation | Test |
|---|---|---|---|---|---|---|---|---|---|
| ACM | 3,025 | Paper-Author-Paper (PAP)
Paper-Subject-Paper (PSP) | 26,416
2,197,556 | 38.08
99.05 | 1,902 | 3 | 600 | 300 | 2,125 |
| DBLP | 2,957 | Author-Paper-Author (APA)
Author-Paper-Conference-Paper-Author (APCPA) | 2,398
1,460,724 | 0
99.82 | 334 | 4 | 600 | 300 | 2,057 |
| Yelp | 2,614 | Business-User-Business (BUB)
Business-Service-Business (BSB)
Business-Rating Levels-Business (BLB) | 525,718
2,475,108
1,484,692 | 83.12
97.49
93.07 | 82 | 3 | 300 | 300 | 2,014 |
| MAG | 113,919 | Paper-Paper (PP)
Paper-Author-Paper (PAP) | 1,806,596
10,067,799 | 64.59
93.48 | 128 | 4 | 40,000 | 10,000 | 63,919 |

## E.2  Hyper-parameters Settings and Infrastructure

We implement all experiments on the platform with PyTorch 1.10.1 and DGL 0.9.1 using an Intel(R) Xeon(R) Platinum 8457C 20 vCPU and an L20 48GB GPU. We perform 5 runs of all experiments and report the average results. In the large MAG data set, InfoMGF-RA takes 80 minutes to complete 5 runs, whereas, on other datasets, both versions of InfoMGF require less than 5 minutes.

Our model is trained with the Adam optimizer, and Table 6 presents the hyper-parameter settings on all datasets. Here, $E$ represents the number of epochs for training, and $lr$ denotes the learning rate. The hidden-layer dimension $d_h$ and representation dimension $d$ of graph encoder GCN are tuned from $\{32, 64, 128, 256\}$. The number of neighbors $k$ for $k$NN is searched from $\{5, 10, 15, 20, 30\}$. The order of graph aggregation $r$ and the number of layers $L$ in GCN are set to 2 or 3, aligning with

Table 6: Details of the hyper-parameters settings.

| Dataset | $E$ | $lr$ | $d_h$ | $d$ | $k$ | $r$ | $L$ | $\rho$ | $\tau_c$ | Random Aug. $\rho_s$ | Generative Aug. $lr_{gen}$ | $\tau$ | $\lambda$ |
|---------|-----|------|-------|-----|-----|-----|-----|--------|----------|------|-------|-----|------|
| ACM  | 100 | 0.01  | 128 | 64 | 15 | 2 | 2 | 0.5 | 0.2 | 0.5 | 0.001 | 1 | 0.01 |
| DBLP | 100 | 0.01  | 64  | 32 | 10 | 2 | 2 | 0.5 | 0.2 | 0.5 | 0.001 | 1 | 1 |
| Yelp | 100 | 0.001 | 128 | 64 | 15 | 2 | 2 | 0.5 | 0.2 | 0.5 | 0.001 | 1 | 1 |
| MAG  | 200 | 0.005 | 256 | 64 | 15 | 3 | 3 | 0   | 0.2 | 0.5 | -     | - | - |

the common layer count of GNN models [69]. The probability $\rho$ of random feature masking is set to 0.5 or 0, and the temperature parameter $\tau_c$ in contrastive loss is fixed at 0.2. For InfoMGF-RA using random graph augmentation, the probability $\rho_s$ of random edge dropping is fixed at 0.5. For InfoMGF-LA with learnable generative graph augmentation, the generator's learning rate $lr_{gen}$ is fixed at 0.001, the temperature parameter $\tau$ in Gumbel-Max is set to 1, and the hyper-parameter $\lambda$ controlling the minimization of mutual information is fine-tuned from $\{0.001, 0.01, 0.1, 1, 10\}$. For the large dataset MAG, we compute the contrastive loss for estimating mutual information in batches, with a batch size of 2560.

# F   Additional Experiments

## F.1   Sensitivity Analysis

We analyze the impact of two important hyper-parameters: the number of neighbors $k$ in $k$NN and hyper-parameter $\lambda$ controlling the influence of mutual information minimization to generate augmented graphs. The performance change of InfoMGF-LA with respect to $k$ is illustrated in Figure 6a. Overall, InfoMGF shows low sensitivity to changes in $k$. The model achieves optimal performance when $k$ is set to 10 or 15. However, when $k$ is very small ($k = 5$), detrimental effects may arise, possibly due to the limited number of beneficial neighbors. As $k$ increases, the performance can still be maintained high. Figure 6b shows the results to $\lambda$ from $\{0.001, 0.01, 0.1, 1, 10\}$. Our proposed model shows low sensitivity to changes in $\lambda$ in general, while the $\lambda$ corresponding to achieving the best performance varies across different datasets.

## F.2   Graph Visualization

Figures 7 and 8, respectively, present the visualizations of the subgraph adjacency matrices of the original multiplex graphs and the learned fused graph $G^s$ on the DBLP and Yelp datasets. In DBLP, the two categories are machine learning (C1) and information retrieval (C2), while in Yelp, the categories are Mexican flavor (C1) and hamburger type (C2). It can be observed that $G^s$ not only removes the inter-class edges in the original structure but also retains key intra-class edges with weights, not just the shared edges. This further demonstrates the effectiveness of InfoMGF in eliminating task-irrelevant noise while preserving sufficient task-relevant information.

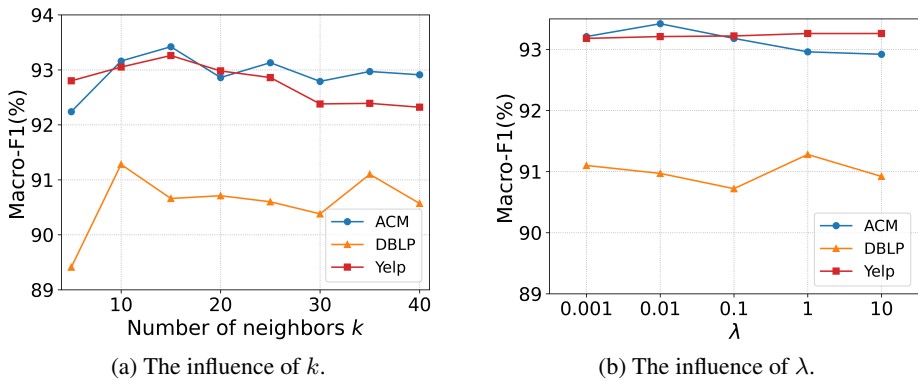

(a) The influence of $k$.    (b) The influence of $\lambda$.

Figure 6: Additional experiments on sensitivity analysis.

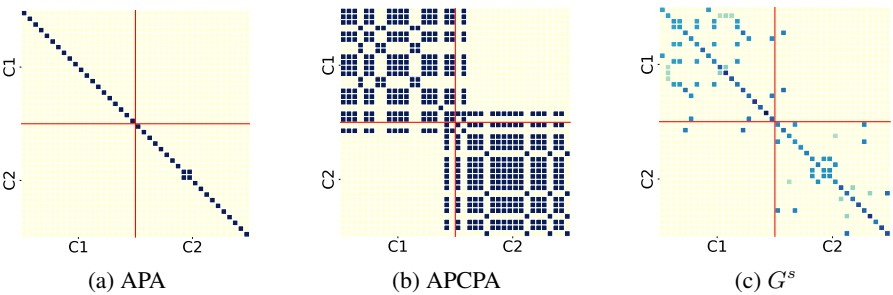

(a) APA      (b) APCPA      (c) $G^s$

Figure 7: Heatmaps of the subgraph adjacency matrices of the original and learned graphs on DBLP.

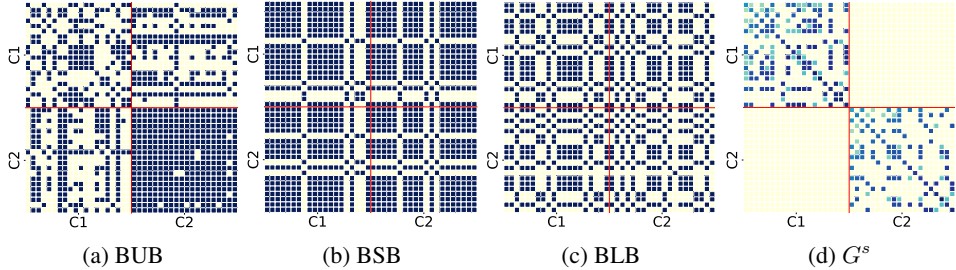

(a) BUB      (b) BSB      (c) BLB      (d) $G^s$

Figure 8: Heatmaps of the subgraph adjacency matrices of the original and learned graphs on Yelp.

