# OpenReview forum: "Beyond Redundancy: Information-aware Unsupervised Multiplex Graph Structure Learning"
_NeurIPS.cc/2024/Conference — NeurIPS 2024 poster_

### Official Review · Reviewer_gjsJ · 2024-07-06

**Soundness:** 3
**Presentation:** 3
**Contribution:** 3
**Rating:** 7
**Confidence:** 4

**Summary:**

- This paper presents an information theory approach to obtain a single graph fused from a multiplex graph, which preserves
   - sufficient task-relevant information
   - while removing task-irrelevant noise.
- A learnable graph augmentation strategy is also developed.
   - The learned graph and representation can be applied to different types of tasks.
- The effectiveness is supported by extensive experimental results.

**Strengths:**

- This paper is well-motivated.
   - The authors find that each graph contains much unique task-relevant information, which is ignored by mainstream contrastive learning-based methods.
- This paper develops multiple graphs non-redundancy principle, which lays the foundation for multiplex graph data process.
   - Two random and generative graph augmentation strategies are accordingly built to capture view-unique task information.
- The experimental results are promising.
   - The framework demonstrates a clear advantage over existing methods, including advanced supervised approaches, highlighting its potential for broad application.
- This paper provides the code and all experimental settings for reproducing the results.

**Weaknesses:**

- The difference between the existing non-redundancy principle and multiplex graph non-redundancy is unclear. Please clarify it.
- The proposed InfoMGF-LA runs out-of-memory on MAG data. The reason should be given.
- It is possible that the proposed method cannot handle real-world large-scale graph. It should be addressed in the future and discussed in the conclusion part.
- The difference between the proposed method and DGM is unclear.

**Questions:**

I list them in **Weaknesses**.

**Limitations:**

Some imitations are addressed in $\S5$.

---

> ### Author Rebuttal · Authors · 2024-08-06
>
> We appreciate your thoughtful feedback. Your constructive criticism is invaluable in refining our work. Below, we give point-by-point responses to your comments.
>
> 1. **The difference between the existing non-redundancy principle and multiplex graph non-redundancy is unclear. Please clarify it.**
>
> The primary distinction between multiplex graph non-redundancy and existing non-redundancy principles lies in our consideration of the complexity and uniqueness of graph-structured data. Existing work mainly focuses on non-redundancy in bimodal/view data of images and texts [1], considering that different views may have unique task-relevant information. For image and text data, the non-redundancy is intuitively easy to understand and verify. However, real-world graph data is often complex and not intuitive, making the analysis of non-redundancy very tricky and difficult. Therefore, we propose the concept of "The Unique Relevant Edge Ratio" and conduct empirical research, creatively explaining the non-redundant properties of graph data from a structural connectivity perspective. Furthermore, this also demonstrates that relying solely on decoupled node representation learning is insufficient for capturing both shared and task-specific information in multiplex graphs; graph structure refinement or relearning is necessary.
>
> [1] Liang et al. "Factorized Contrastive Learning: Going beyond Multi-view Redundancy." NeurIPS (2023).
>
> 2. **The proposed InfoMGF-LA runs out-of-memory on MAG data. The reason should be given.**
>
>  As a large-scale real-world multiplex graph dataset, MAG’s PAP graph contains a substantial number of edges—10,067,799 to be precise. The primary reason InfoMGF cannot be successfully trained lies in its use of learnable generative augmentation, which necessitates probabilistic modeling of each edge in the original graphs (Equation 4 of our paper). During model optimization, the weight of each edge contributes to gradient calculations.  The significant memory requirements render InfoMGF-LA impractical for training on MAG data. Despite this limitation, our alternative approach, InfoMGF-RA, still achieves remarkable performance. We believe that this outcome underscores the efficacy of our method.
>
> 3. **It is possible that the proposed method cannot handle real-world large-scale graph. It should be addressed in the future and discussed in the conclusion part.**
>
> We appreciate your concern regarding the scalability of our method. We will include a discussion on scalability in Section 5 “Conclusion and Limitation,” which will make our contributions and limitations more complete and accurate. To enable InfoMGF-LA to handle real-world large-scale graph data successfully, we will focus on developing scalable and learnable graph augmentation methods to replace the existing edge-wise probability modeling augmentation. According to current research [2], most deep graph generation methods still cannot escape the complexity of $\mathcal{O}(N^2)$ or $\mathcal{O}(\vert E\vert)$, making them unsuitable for large-scale graph data. This issue becomes even more challenging when considering node attribute information. In the future, we may explore using diffusion-based generative models to address large graph augmentation or generation.
>
> [2] Zhu et al. "A Survey on Deep Graph Generation: Methods and applications." LOG (2022).
>
> 4. **The difference between the proposed method and DGM is unclear.**
>
> The main difference between InfoMGF and DMG [3] lies in the task objectives and potential applicability. Specifically, we aim to address the reliability of multiplex graph structures by leveraging unsupervised graph structure learning to preserve task-relevant information while removing task-irrelevant noise. In contrast, DMG is a graph-fixed method that focuses solely on node representation learning. This limitation prevents DMG, and similar graph-fixed methods, from effectively handling noisy real-world graph data. InfoMGF, as a data-centric GSL framework, can achieve both graph structure refinement and representation learning, highlighting its broader range of potential applications.
>
> [3] Mo et al. "Disentangled Multiplex Graph Representation Learning." ICML (2023).
>
> Once again, we sincerely appreciate your time and effort in reviewing our paper. Your constructive criticism has been invaluable in refining our work, and we hope these adjustments and explanations can address your concerns satisfactorily.

---

> > ### Author Response · Authors · 2024-08-14
> > **Thanks for the raising**
> >
> > We are grateful to Reviewer gjsJ for suggesting this improvement! If you have any additional questions or concerns, please feel free to discuss them with us.

---

### Official Review · Reviewer_ei6T · 2024-07-06

**Soundness:** 3
**Presentation:** 3
**Contribution:** 3
**Rating:** 7
**Confidence:** 5

**Summary:**

The paper introduces InfoMGF (Information-aware Unsupervised Multiplex Graph Fusion), a novel framework aimed at addressing the issue of graph structure reliability in Multiplex Graphs. The primary goal is to refine graph structures to eliminate noise and maximize task-relevant information. Theoretical analysis and comprehensive experimental results validate its effectiveness.

**Strengths:**

1.	Originality: The paper addresses a critical gap in Unsupervised Multiplex Graph Learning (UMGL) by focusing on the reliability of graph structures, which is often overlooked in existing research.
2.	Quality: The proposed InfoMGF framework effectively refines graph structures to eliminate noise and maximizes both view-shared and view-unique task-relevant information. Theoretical analyses provided in the paper validate the effectiveness of InfoMGF in capturing task-relevant information and improving graph fusion. Extensive experiments demonstrate that InfoMGF outperforms various baselines and even sophisticated supervised approaches in different downstream tasks.
3.	Clarity: The paper is generally clearly written and well organized.

**Weaknesses:**

1.	Scalability: The framework involves several steps. Though the paper provides the complexity analysis in Appendix for each step, it is still unclear what is the overall complexity.
2.	Reproducibility: The authors share the code for reproducibility. However, I didn’t see the datasets.
3.	Accuracy: The authors should check for the few grammatical and spelling errors that occur in the text.

**Questions:**

As above.

**Limitations:**

Yes.

---

> ### Author Rebuttal · Authors · 2024-08-06
>
> We appreciate your thoughtful feedback. Your constructive criticism is invaluable in refining our work. Below, we give point-by-point responses to your comments.
>
> 1. **Scalability: The framework involves several steps. Though the paper provides the complexity analysis in Appendix for each step, it is still unclear what is the overall complexity.**
>
>  We apologize for the unclear and incomplete description of the time complexity. We will revise the time complexity analysis and include the overall complexity. Here is the revised version:
>
> We first analyze the time complexity of each component in InfoMGF. In this paragraph, let $V$, $N$, and $m$ represent the numbers of graphs, nodes, and edges, while $b_1$ and $b_2$ denote the batch sizes of locality-sensitive $k$NN and contrastive loss computation. The layer numbers of graph learner, graph encoder GCN, and non-linear projector are denoted as $L_1$, $L_2$, and $L_3$, respectively. The feature, hidden layer, and representation dimensions are denoted as $d_f$, $d_h$, and $d$, respectively. We analyze the complexity of $k$NN and GCN in scalable versions. Before training, scalable SGC is applied with a complexity of $\mathcal{O}(Vmrd_f)$ related to the aggregation order $r$. During training, we first conduct a graph learner with scalable $k$NN requiring $\mathcal{O}(VNL_1d_f+VNb_1d_f)$. For the GCN encoder and non-linear projector, the total complexity is $\mathcal{O}\left(VmL_2d_h+Vmd+VNL_2d_h^2+VNd_h(d+d_f)+VNL_3d^2\right)$. Within the graph augmentation module, the complexity of feature masking is $\mathcal{O}(Nd_f)$. The learnable generative graph augmentation in InfoMGF-LA has a complexity of $\mathcal{O}(VNd_fd_h+Vmd_h+VNd_fd)$, where the first two terms are contributed by augmented graph generator, and the last one is for the decoder. For InfoMGF-RA, the random edge dropping requires $\mathcal{O}(Vm)$ time complexity. For the loss computation, the complexity is $\mathcal{O}(V^2Nb_2d)$.
>
> To simplify the overall complexity, we denote the larger terms within $L_1$, $L_2$, and $L_3$ as $L$, the larger terms between $d_h$ and $d$ as $\hat{d}$, the larger terms between $b_1$ and $b_2$ as $B$. Since the scalable SGC operation only needs to be performed once before training, its impact on training time is negligible. Therefore, we only consider the total complexity during the training process. The overall complexity of both InfoMGF-RA and InfoMGF-LA is: $\mathcal{O}(VmL\hat{d}+VNL\hat{d}^2+VNd_f(\hat{d}+L)+VNB(d_f+V\hat{d}))$, which is comparable to mainstream unsupervised GSL models, including our baselines. For example, SUBLIME [1] needs to be trained on each graph in a multiplex graph dataset, and its time complexity is: $\mathcal{O}(VmL\hat{d}+VNL\hat{d}^2+VNd_f(\hat{d}+L)+VNB(d_f+\hat{d}))$, which only has a slight difference in the last term compared to the time complexity of our method.
>
> [1] Liu et al. "Towards Unsupervised Deep Graph Structure Learning." WWW (2022).
>
>
>
> 2. **Reproducibility: The authors share the code for reproducibility. However, I didn’t see the datasets.**
>
> At the end of the abstract, there is a link to an Anonymous GitHub repository that contains the complete code for our method. All the used multiplex graph datasets are publicly available, and we have included the corresponding references for each dataset. We will release all the datasets after acceptance and include them along with the complete code in the GitHub repository.
>
> 3. **Accuracy: The authors should check for the few grammatical and spelling errors that occur in the text.**
>
> We will carefully review and correct any grammar and spelling errors in the manuscript. We focus on improving the clarity and coherence of our writing while ensuring language accuracy.
>
> Once again, we sincerely appreciate your time and effort in reviewing our paper. Your constructive criticism has been invaluable in refining our work, and we hope these adjustments and explanations could address your concerns satisfactorily.

---

> > ### Comment · Reviewer_ei6T · 2024-08-12
> >
> > Thanks to the author for the reply, I have no doubts anymore and I will improve my score.

---

> > > ### Author Response · Authors · 2024-08-13
> > > **Thanks for the raising**
> > >
> > > We are grateful to Reviewer ei6T for suggesting this improvement! If you have any additional questions or concerns, please feel free to discuss them with us.

---

### Official Review · Reviewer_cF8g · 2024-07-11

**Soundness:** 3
**Presentation:** 3
**Contribution:** 3
**Rating:** 7
**Confidence:** 4

**Summary:**

The paper introduces InfoMGF, an innovative framework for Unsupervised Multiplex Graph Learning (UMGL) that addresses the often-overlooked issue of graph structure reliability. InfoMGF refines graph structures by removing task-irrelevant noise and maximizing task-relevant information through mutual information maximization. Extensive experiments demonstrate its superior performance over various baselines and even some supervised methods, validating its effectiveness in enhancing node representation learning.

**Strengths:**

- New Problem Formulation: The paper pioneers the investigation of graph structure reliability in multiplex graphs, which is a significant advancement in the field. Multiplex graphs enrich the representation of real-world systems and its analysis is very difficult inherently.
- Theoretical Analysis: The several theorems are quite interesting and provide a solid foundation for the proposed method. In particular, Theorem 3 proves the necessity of fusing multiplex graphs.
- Extensive Evaluation: The framework is thoroughly tested against various state-of-the-art methods on both node clustering and classification tasks, showcasing its robustness and effectiveness across different tasks. The comparison methods are representative and new.

**Weaknesses:**

- Robustness: Fig.4 shows that the proposed method is very robust to structure noise. However, more analysis is needed. Both InfoMGF and SUBLIME are structure learning methods. Compared to InfoMGF，Why does the performance of SUBLIME degrade rapidly in the case of edge deletions?
- Clarity: The paper develops two algorithms in this paper: InfoMGF-RA and InfoMGF-LA. However, it is a little confusion that what is the difference in their objective functions.

**Questions:**

1.	There are some small errors in Algorithm 1. In particular, the title of it is InfoMGF-LA, however, line 11 also includes the operation for InfoMGF-RA.
2.	The proposed method depends on the assumption of optimal augmentation. How to guarantee that the used feature and structure augmentations are optimal? It is still unclear to me.
3.	The authors discuss the robustness against structure noise. How about feature noise? Could you share your intuition on this matter?

**Limitations:**

The paper discusses the limitations.

---

> ### Author Rebuttal · Authors · 2024-08-06
>
> We appreciate your thoughtful feedback. Your constructive criticism is invaluable in refining our work. Below, we give point-by-point responses to your comments.
>
> **W1. Fig.4 shows that the proposed method is very robust to structure noise. However, more analysis is needed. Both InfoMGF and SUBLIME are structure learning methods. Compared to InfoMGF. Why does the performance of SUBLIME degrade rapidly in the case of edge deletions?**
>
> As you can see in Fig.4, our model is significantly more robust than SUBLIME when deleting edges. This can be attributed to the following factors:
>
> 1. When edges are deleted, the task-relevant information contained in the graph tends to decrease. SUBLIME, as a single graph method, cannot fully leverage the task-relevant information from different graphs. In contrast, InfoMGF considers multiplex graph non-redundancy, enabling it to capture both shared and unique task-relevant information from all graphs. Consequently, when the edge deletion ratio is substantial, InfoMGF outperforms SUBLIME significantly.
> 2. Fig.4 shows the robustness comparison between InfoMGF-LA and other methods. Compared to existing methods (including SUBLIME) that use random graph augmentation, the novel learnable generative augmentation used in InfoMGF-LA is more reliable and interpretable. This approach retains more task-relevant information while reducing irrelevant noise. As a result, our method is more robust across various scenarios.
>
> **W2 & Q1. The paper develops two algorithms in this paper: InfoMGF-RA and InfoMGF-LA. However, it is a little confusion that what is the difference in their objective functions. There are some small errors in Algorithm 1. In particular, the title of it is InfoMGF-LA, however, line 11 also includes the operation for InfoMGF-RA.**
>
> We apologize for any confusion. InfoMGF-RA and InfoMGF-LA have the same optimization objective (Equation 10), with the primary difference lying in the generation of augmented graphs. Graph augmentation includes both feature and structure augmentation. Notably, both RA and LA use simple yet effective random feature masking for feature augmentation. However, for structure augmentation, RA and LA employ random edge dropping and learnable generative augmentation, respectively. For InfoMGF-RA, both feature and structure augmentations are non-parametric methods, so its overall objective is Equation 10. In contrast, InfoMGF-LA requires to train the learnable generative augmentation module (optimized by Equation 11). Therefore, we propose an **alternating optimization strategy** to iteratively optimize the total model loss ($\mathcal{L}$) and the augmented graph generator loss ($\mathcal{L}_{gen}$), which is described in lines 233-237 of our paper.
>
> Regarding your question about Algorithm 1, there seems to be a misunderstanding. Since InfoMGF-LA also requires random feature masking to generate augmented features, the description in Algorithm 1 is accurate. To make the distinction between InfoMGF-RA and InfoMGF-LA clearer, we will include the algorithm for InfoMGF-RA in the final version.
>
> **Q2. The proposed method depends on the assumption of optimal augmentation. How to guarantee that the used feature and structure augmentations are optimal?**
>
> In fact, due to the unsupervised setting where label information is unavailable, obtaining rigorously theoretically guaranteed $G_v^\prime$ remains challenging. Therefore, we have to approach optimal augmentation from a practical perspective.
>
> Previous research often uses random augmentation, assuming that most of the perturbed information is task-irrelevant. Additionally, existing methods commonly augment features and structures simultaneously, as both typically contain rich task-relevant information. In contrast, our approach takes further steps:
>
> 1. **Feature Augmentation**: We continue to employ simple yet effective random feature masking. Extensive pretraining studies in graph learning [1], CV [2], and NLP [3] have demonstrated that random masking performs remarkably well when augmenting information-dense data, without requiring intricate techniques.
> 2. **Structure Augmentation**: Random edge dropping may lack reliability and interpretability, so we propose the learnable generative augmentation, which implements edge-wise probability modeling in the original graphs. To effectively train the augmented graph generator, we design the loss function $\mathcal{L}_{gen}$ based on **the principle of optimal augmentation**. The reconstruction term constrains the augmented graph to retain essential information, while the mutual information minimization term aims to reduce irrelevant noise within the augmented graph. We delve into the optimization process and advantages of our proposed augmentation in the relevant section of our paper (lines 183-198).
>
> In summary, due to the unavailability of label information in unsupervised tasks, we cannot achieve rigorously theoretically guaranteed augmentation. Nevertheless, we adhere to the principle of optimal augmentation to develop learnable generative augmentation and its optimization objectives. Extensive experiments further validate the effectiveness of our approach.
>
> [1] "Graph Contrastive Learning with Augmentations." NeurIPS (2020).
>
> [2] "Masked Autoencoders Are Scalable Vision Learners." CVPR (2022).
>
> [3] "UniLMv2: Pseudo-Masked Language Models for Unified Language Model Pre-Training." ICML (2020).
>
>
> **Q3. The authors discuss the robustness against structure noise. How about feature noise? Could you share your intuition on this matter?**
>
> Following your advice, we add some additional experiments on this issue, where the experimental results and analysis are in the PDF file of the Global Author Rebuttal.
>
> Once again, we sincerely appreciate your time and effort in reviewing our paper. Your constructive criticism has been invaluable in refining our work, and we hope these adjustments and explanations can address your concerns satisfactorily.

---

### Official Review · Reviewer_2Joh · 2024-07-12

**Soundness:** 3
**Presentation:** 3
**Contribution:** 3
**Rating:** 6
**Confidence:** 4

**Summary:**

The authors develop a novel approach to improve Unsupervised Multiplex Graph Learning by refining graph structures to eliminate noise and maximize relevant information. The method utilizes mutual information maximization to integrate multiple graph views effectively. Theoretical validation and comprehensive experiments show that the proposed method outperforms existing methods.

**Strengths:**

1.	Multiplex graph provides an efficient representation of complex systems. This paper focuses on non-redundancy issue, which is a new perspective and opens up a new avenue for future research.
2.	The proposed method adopts an unsupervised and generalized approach. Its performance surpasses several supervised approaches, underscoring its potential for practical applications.
3.	The framework’s performance is validated through comprehensive experiments and compared with more than 20 methods.
4.	Visualization is also a strong point of this paper. The figures of node correlation, heatmaps of the subgraph, and unique relevant edge ratio are very illustrative.

**Weaknesses:**

1.	According to Table 1 and 2, it seems that the proposed method improves more on clustering than classification.
2.	Overall, this paper is well-organized. However, the writing could be improved in terms of tone and words.
3.	There are too many notations, which are confusing.

**Questions:**

1.	Is there any explanation about why the method performs better on clustering than classification?
2.	How to solve the above issue?
3.	The font of k in the caption of fig.5a is not correct.
4.	In the Appendix, the authors proof Proposition 1, however, there is no corresponding one in the main paper.

---

> ### Author Rebuttal · Authors · 2024-08-06
>
> We appreciate your thoughtful feedback. Your constructive criticism is invaluable in refining our work. Below, we give point-by-point responses to your comments.
>
> **Q1 & W1 & Q2. According to Table 1 and 2, it seems that the proposed method improves more on clustering than classification. Is there any explanation about why the method performs better on clustering than classification? How to solve the above issue?**
>
> Thanks for your concern. Your observations are accurate and well-founded. The bigger improvement of our method in clustering  compared to classification can be attributed to the following factors:
>
> 1. **Differences in Baselines**: In the clustering task, we primarily compare our method against unsupervised multiplex graph learning (UMGL) methods. As summarized in the Introduction, existing UMGL methods are graph-fixed, which means they struggle to handle real-world noisy graph data. In contrast, for a comprehensive evaluation, we compare our method against a wide range of supervised/unsupervised single graph structure learning (GSL) methods in the classification task. These GSL methods refine the graph structure, leading to better baseline performances. This partially explains the substantial improvement in clustering compared to classification.
> 2. **Incorporating Pre-trained Knowledge**: In the clustering task, we evaluate the node representations $Z$ of the learned fused graph by K-means. Notably, $Z$ not only contains information from the learned fused graph but also incorporates ample pre-trained knowledge from the GCN encoder parameters. This pre-trained knowledge contains more underlying information inside multiplex graph data, thereby enhancing node clustering performance. However, in the classification task, we retrain a new GCN on the learned fused graph to align with existing GSL paradigms, which means that the classification performance reflects only the quality of the learned graph and lacks the richer pre-trained knowledge. This also explains why the improvement in classification is smaller compared to clustering.
>
> **Future Approaches**: In the future, we will leverage transfer learning to migrate pre-trained knowledge to a broader range of downstream tasks, including classification. For instance, we can initialize the classifier for classification tasks using pre-trained GCN parameters and fine-tune it to improve the performance. Additionally, we will also explore cross-domain transfer learning, enhancing our method’s out-of-distribution detection and generalization capabilities for handling real-world graph data. Furthermore, we will delve into multiplex graph structure learning in supervised or semi-supervised scenarios. By leveraging annotated information, we aim to improve the quality of the learned graphs.
>
>
>
> **W3. There are too many notations,  which are confusing.**
>
> Sorry for the trouble caused by our symbols, we agree that too many notations would confuse the readers. Hence, we summarize frequently used notations in our paper as a **Table** in the PDF file of the Global Author Rebuttal. Meanwhile, we also explain some important symbols in the following:
>
> 1. **Graph symbols.** Such as $G_v=\\{A_v,X\\}$, $G^s_v=\\{A_v^s,X\\}$, and $G^{\prime}_v=\\{A_v^{\prime},X^{\prime} \\}$ to denote the original, refined, and augmented $v$-th graph respectively. The learned fused graph is denoted as $G^s=\\{A^s,X\\}$. It is worth noting that $A^s$ and all $A^s_v$ are generated via the graph learner.
> 2. **Vectors and matrices used in graph learning.** Most notations appear in the graph learning stage, for instance, we use $X_i^{v}$ to denote the view-specific feature of $i$-th node in $v$-th graph. $H^v$ is the node embeddings of the $v$-th original graph in graph learner, which is used to generate the refined graph. $Z^{v}=\mathrm{GCN}(A_{v}^{s}, X)$ is the node representations of the $v$-th refined graph in the GCN encoder, which is used for the loss calculation. It is obvious that these notations cannot be omitted for their diverse utilizes.
> 3. **Units for optimization.** During optimization, we introduce the mutual information lower/upper bound and loss function, which combine the aforementioned notations. Superscripts and subscripts are important for distinguishing different views and different nodes.
>
> We will include the “Frequently Used Notations” table and explanations of the symbols in the Appendix. Thanks for your constructive suggestion.
>
>
>
> **Q3.  The font of k in the caption of fig.5a is not correct.**
>
> Sorry for the mistake we made. We will make corrections on this issue in the final version of our paper. And we will conduct a thorough review of the paper to ensure there are no typographical or language errors.
>
>
>
> **Q4. In the Appendix, the authors proof Proposition 1, however, there is no corresponding one in the main paper.**
>
> Thank you for your valuable feedback. We will incorporate the corresponding proposition into the main paper, which will make our theoretical focus clearer and more prominent.
>
> Once again, we sincerely appreciate your time and effort in reviewing our paper. Your constructive criticism has been invaluable in refining our work, and we are more than happy to add clarifications to address any additional recommendations and reviews from you!

---

### Author Rebuttal · Authors · 2024-08-06

We sincerely thank all the reviewers for their valuable and insightful comments. We are glad that the reviewers find that our studied problem is novel and significant. Here, we provide a PDF file to further address the reviewers’ concerns regarding the clarity of the paper and the completeness of the experiments.

**P1: The robustness analysis against feature noise. (To Reviewer cF8g)**

 Figure 5 in the PDF file shows the performance of InfoMGF and various baselines on the ACM dataset when injecting random feature noise. It can be observed that InfoMGF exhibits excellent robustness against feature noise, while the performance of SUBLIME degrades rapidly. As a single graph structure learning method, SUBLIME’s performance heavily relies on the quality of node features. In contrast, our method can directly optimize task-relevant information in multi-view graph structures (e.g., edges shared across multiple graphs are likely to be shared task-relevant information, which can be directly learned through $\mathcal{L}_s$), thereby reducing dependence on node features. Consequently, InfoMGF demonstrates superior robustness against feature noise.

**P2: The algorithmic process for InfoMGF-RA. (To Reviewer cF8g)**

Thank you for your question about the specific differences between InfoMGF-RA and InfoMGF-LA. In the PDF file, Algorithm 1 provides the algorithmic process for InfoMGF-RA. You can compare it with the algorithmic process for InfoMGF-LA in the Appendix of our original paper for a better understanding of their differences. Specifically, the total loss for both is the same, with the main difference lying in the graph structure augmentation methods. Additionally, InfoMGF-LA employs an alternating optimization strategy, which involves two optimization passes per epoch to optimize both the total loss and the augmented graph generator loss. In contrast, the optimization process for InfoMGF-RA is simpler, requiring only the optimization of the total loss.

**P3: Detailed explanations of frequently used notations. (To Reviewer 2Joh)**

Thank you for your suggestions regarding the notation definitions. To make the symbols used in the paper clearer and easier to understand, we have provided explanations in Table 4 of the PDF file under “Frequently used notations”. We will also include this table in the Appendix of the final version.

**P4: The scalability of InfoMGF. (To Reviewers ei6T and gjsJ)**

Thank you for your concerns about scalability. Here, we provide the overall complexity of our method. Let $V$, $N$, $m$, and $d_f$ represent the numbers of graphs, nodes, edges, and features. To simplify the overall complexity, we denote the larger terms within the layers of graph learner, graph encoder GCN, and non-linear projector as $L$, the larger terms between hidden-layer dimension and representation dimension as $\hat{d}$, the larger terms between batch sizes of locality-sensitive $k$NN and contrastive loss as $B$. The overall complexity of both InfoMGF-RA and InfoMGF-LA during the training process is $\mathcal{O}(VmL\hat{d}+VNL\hat{d}^2+VNd_f(\hat{d}+L)+VNB(d_f+V\hat{d}))$, which is comparable to mainstream unsupervised GSL models, including our baselines. For example, SUBLIME [1] needs to be trained on each graph in a multiplex graph dataset, and its time complexity is $\mathcal{O}(VmL\hat{d}+VNL\hat{d}^2+VNd_f(\hat{d}+L)+VNB(d_f+\hat{d}))$, which only has a slight difference in the last term compared to the time complexity of our method.

[1] Liu et al. "Towards Unsupervised Deep Graph Structure Learning." WWW (2022).

---

### Comment · Area_Chair_UjAt · 2024-08-11
**Reminder to Check Authors' Reply**

Dear Reviewers,

Would you mind checking the authors' responses?

- AC

---

### Decision · Program_Chairs · 2024-09-25

**Decision:**

Accept (poster)

**Comment:**

This paper introduces InfoMGF, a framework for unsupervised multiplex graph learning that refines graph structures to remove noise and maximize task-relevant information through mutual information maximization across different graph views. The method effectively addresses the issue of graph structure reliability and shows superior performance in clustering and classification tasks, even outperforming some supervised approaches. While initial concerns were raised about scalability and the clarity of differences between InfoMGF variants and other methods, the authors' rebuttal provided detailed explanations and additional experiments that addressed these concerns.